# Insights into early animal evolution from the genome of the xenacoelomorph worm *Xenoturbella bocki*

Philipp H Schiffer[1,2]*, Paschalis Natsidis[1], Daniel J Leite[1,3], Helen E Robertson[1], François Lapraz[1,4], Ferdinand Marlétaz[1], Bastian Fromm[5], Liam Baudry[6], Fraser Simpson[1], Eirik Høye[7,8], Anne C Zakrzewski[1,9], Paschalia Kapli[1], Katharina J Hoff[10,11], Steven Müller[1,12], Martial Marbouty[13], Heather Marlow[14], Richard R Copley[15], Romain Koszul[13], Peter Sarkies[16], Maximilian J Telford[1]*

[1]Center for Life's Origins and Evolution, Department of Genetics, Evolution and Environment, University College London, London, United Kingdom; [2]worm~lab, Institute of Zoology, University of Cologne, Cologne, Germany; [3]Department of Biosciences, Durham University, Durham, United Kingdom; [4]Université Côte D'Azur, CNRS, Inserm, iBV, Nice, France; [5]The Arctic University Museum of Norway, UiT – The Arctic University of Norway, Tromsø, Norway; [6]Collège Doctoral, Sorbonne Université, Paris, France; [7]Department of Tumor Biology, Institute for Cancer Research, The Norwegian Radium Hospital, Oslo University Hospital, Oslo, Norway; [8]Institute of Clinical Medicine, Medical Faculty, University of Oslo, Oslo, Norway; [9]Museum für Naturkunde, Leibniz Institute for Evolution and Biodiversity Science, Berlin, Germany; [10]University of Greifswald, Institute for Mathematics and Computer Science, Greifswald, Germany; [11]University of Greifswald, Center for Functional Genomics of Microbes, Greifswald, Germany; [12]Royal Brompton Hospital, Guy's and St Thomas' NHS Foundation Trust, London, United Kingdom; [13]Institut Pasteur, Université de Paris, CNRS UMR3525, Unité Régulation Spatiale des Génomes, Paris, France; [14]The University of Chicago, Division of Biological Sciences, Chicago, United States; [15]Laboratoire de Biologie du Développement de Villefranche-sur-mer (LBDV), Sorbonne Universite, Villefranche-sur-mer, France; [16]Department of Biochemistry, University of Oxford, Oxford, United Kingdom

*For correspondence:
philipp.schiffer@gmail.com (PHS);
m.telford@ucl.ac.uk (MJT)

**Competing interest:** The authors declare that no competing interests exist.

**Abstract** The evolutionary origins of Bilateria remain enigmatic. One of the more enduring proposals highlights similarities between a cnidarian-like planula larva and simple acoel-like flatworms. This idea is based in part on the view of the Xenacoelomorpha as an outgroup to all other bilaterians which are themselves designated the Nephrozoa (protostomes and deuterostomes). Genome data can provide important comparative data and help understand the evolution and biology of enigmatic species better. Here, we assemble and analyze the genome of the simple, marine xenacoelomorph *Xenoturbella bocki*, a key species for our understanding of early bilaterian evolution. Our highly contiguous genome assembly of *X. bocki* has a size of ~111 Mbp in 18 chromosome-like scaffolds, with repeat content and intron, exon, and intergenic space comparable to other bilaterian invertebrates. We find *X. bocki* to have a similar number of genes to other bilaterians and to have retained ancestral metazoan synteny. Key bilaterian signaling pathways are also largely complete and most bilaterian miRNAs are present. Overall, we conclude that *X. bocki* has a complex genome typical of bilaterians, which does not reflect the apparent simplicity of its body plan that has been so important to proposals that the Xenacoelomorpha are the simple sister group of the rest of the Bilateria.

## Editor's evaluation

The authors provide a high-quality genome of the xenacoelomorph worm *Xenoturbella bocki* and discuss its structure and evolution. Understanding the genomic structure of this group provides important insights into bilaterian evolution. The authors make a solid case that the data they present is consistent with Xenacoelomorpha being a secondarily simplified member of Deuterostomia rather than a primitively simple sister group to all other bilaterians.

## Introduction

*Xenoturbella bocki* (*Figure 1*) is a morphologically simple marine worm first described from specimens collected from muddy sediments in the Gullmarsfjord on the west coast of Sweden. There are now six described species of *Xenoturbella* – the only genus in the higher-level taxon of Xenoturbellida (*Telford, 2008*). *X. bocki* was initially included as a species within the Platyhelminthes (*Westblad, 1949*), but molecular phylogenetic studies have shown that Xenoturbellida is the sister group of the Acoelomorpha, a second clade of morphologically simple worms also originally considered Platyhelminthes: Xenoturbellida and Acoelomorpha constitute their own phylum, the Xenacoelomorpha (*Philippe et al., 2019*; *Cannon et al., 2016*). In addition to multiple phylogenetic studies that support the monophyly of the phylum, Xenacoelomorpha is convincingly supported by classical analysis in the field of evolution of development, for example, their sharing unique amino acid signatures in their Caudal genes (*Philippe et al., 2019*) and a Hox4/5/6 gene (*Ueki et al., 2019*). Here we analyze our data in this phylogenetic framework of a monophyletic taxon.

The simplicity of xenacoelomorph species compared to other bilaterians is a central feature of discussions over their evolution. While Xenacoelomorpha are clearly monophyletic, their phylogenetic position within the Metazoa has been controversial for a quarter of a century. There are two broadly discussed scenarios: a majority of studies have supported a position for Xenacoelomorpha as the sister group of all other Bilateria (the Protostomia and Deuterostomia, collectively named Nephrozoa) (*Jimenez-Guri et al., 2006*; *Ryan et al., 2006*; *Jékely, 2013*); work we have contributed to *Telford, 2008*; *Philippe et al., 2019*; *Philippe et al., 2011*; *Bourlat et al., 2006*, has instead placed Xenacoelomorpha within the Bilateria as the sister group of the Ambulacraria (Hemichordata and Echinodermata) to form a clade called the Xenambulacraria (*Philippe et al., 2011*).

*X. bocki* has neither organized gonads nor a centralized nervous system. It has a blind gut, no body cavities, and lacks nephrocytes (*Nakano, 2015*). If Xenacoelomorpha is the sister group to Nephrozoa, these character absences can be parsimoniously interpreted as representing the primitive state of the Bilateria. According to advocates of the Nephrozoa hypothesis, these and other characters absent in Xenacoelomorpha must have evolved in the lineage leading to Nephrozoa after the divergence of Xenacoelomorpha. More generally, there has been a tendency to interpret Xenacoelomorpha (especially Acoelomorpha) as living approximations of Urbilateria (*Hejnol et al., 2009*; *Hejnol and Martindale, 2008*).

An alternative explanation for the simple body plan of xenaceolomorphs is that it is derived from that of more complex urbilaterian ancestors through loss of morphological characters. The loss or remodeling of morphological complexity is a common feature of evolution in many animal groups and is typically associated with unusual modes of living (*Martynov et al., 2020*; *Westheide, 1987*) – in particular, the adoption of a sessile (sea squirts, barnacles) or parasitic (neodermatan flatworms, orthonectids) lifestyle, extreme miniaturization (e.g., tardigrades, orthonectids), or even neoteny (e.g., flightless hexapods).

The biology of *Xenoturbella* is difficult to study in vivo – they are hard to collect and mostly inactive in culture: knowledge of their embryology is restricted to one descriptive paper of a handful of embryos (*Nakano et al., 2013*). One route to better understanding the biology of this key taxon in the phylogeny of the animals is to read and study their genome.

In the past, some genomic features gleaned from analysis of various Xenacoelomorpha have been used to test these evolutionary hypotheses. For example, the common ancestor of the protostomes and deuterostomes has been reconstructed with approximately eight Hox genes but only four have been found in the Acoelomorpha (Nemertoderma) and five in *Xenoturbella*. This has been interpreted as a primary absence with the full complement of eight proposed to have appeared subsequent to the divergence of Xenacoelomorpha and Nephrozoa. Similarly, analysis of the microRNAs (miRNAs) of

**eLife digest** *Xenoturbella bocki* is a small marine worm predominantly found on the seafloor of fjords along the west coast of Sweden. This simple organism's unusual evolutionary history has long intrigued zoologists as it is not clear how it is related to other animal groups. The worm may belong to one of the earliest branches of the animal kingdom, which would explain its simple body. On the other hand, it could be related to a more complex group, the deuterostomes, which includes a wide range of animals, from mammals and birds to sea urchins and starfish.

Understanding *X. bocki*'s evolution could provide valuable insights into how bilaterians evolved as a whole. Unlike its close relatives, the acoelomorphs, *X. bocki* evolves more slowly, which makes it simpler to study its genome. As a result, it serves as a starting point for investigating the evolutionary processes and genetics underpinning the broader group of bilaterians.

To better understand the evolution of *X. bocki*'s simple body, Schiffer et al. asked whether its genome is simpler or differs in other ways from that of more complex bilaterian organisms. Sequencing the entire *X. bocki* genome revealed that it has a similar number of genes to that of other animals and includes the genes required for complex biochemical pathways**.** Reconstructing the worm's chromosomes – the structures that house genetic information – showed that the *X. bocki* genes are also distributed in a manner similar to those in other animals.

The findings suggest that, despite its simple body plan, *X. bocki* has a complex genome that is typical of bilaterians. This challenges the idea that *X. bocki* belongs to a more primitive, simplified sister group to Bilateria and provides a starting point for further studies of how this simple worm evolved.

an acoelomorph, *Symsagittifera roscoffensis*, found that many bilaterian miRNAs were absent from its genome (*Sempere et al., 2006*). Some of the missing bilaterian miRNAs, however, were subsequently observed in *Xenoturbella* (*Philippe et al., 2011*).

The few xenacoelomorph genomes available to date are from the acoel *Hofstenia miamia* (*Gehrke et al., 2019*) – like other Acoelomorpha it shows accelerated sequence evolution relative to *Xenoturbella* (*Philippe et al., 2019*) – and from two closely related species *Praesagittifera naikaiensis* (*Arimoto et al., 2019*) and *Symsagittifera roscoffensis* (*Martinez et al., 2023*). The analyses of gene content of *Hofstenia* showed similar numbers of genes and gene families to other bilaterians (*Gehrke et al., 2019*), while an analysis of the neuropeptide content concluded that most bilaterian neuropeptides were present in Xenacoelomorpha (*Moroz et al., 2021*).

In order to infer the characteristics of the ancestral xenacoelomorph genome, and to complement the data from the Acoelomorpha, we describe a highly scaffolded genome of the slowly evolving xenacoelomorph *X. bocki*. Our data allow us to contribute knowledge of Xenacoelomorpha and *Xenoturbella* in particular of genomic traits, such as gene content and genome structure and to help reconstruct the genome structure and composition of the ancestral xenacoelomorph. Our data suggest that, while *Xenoturbella* is generally described as having a very simple body (interpreted by many as primitively simple), its genome is of a similar complexity to many other bilaterians, perhaps lending support to the idea that the simplicity of *X. bocki* is derived.

## Results

### Assembly of a draft genome of *X. bocki*

We collected *X. bocki* specimens (*Figure 1*) from the bottom of the Gullmarsfjord close to the biological field station in Kristineberg (Sweden). These adult specimens were starved for several days in tubes with artificial sea water, and then sacrificed in lysis buffer. We extracted high molecular weight (HMW) DNA from single individuals for each of the different sequencing steps below.

We assembled a high-quality draft genome of *X. bocki* using one short read Illumina library and one TruSeq Synthetic Long Reads (TSLR) Illumina library. We used a workflow based on a primary assembly with SPAdes ('Materials and methods'; *Bankevich et al., 2012*). The primary assembly had an N50 of 8.5 kb over 37,880 contigs with a maximum length of 206,709 bp. After using the redundans pipeline

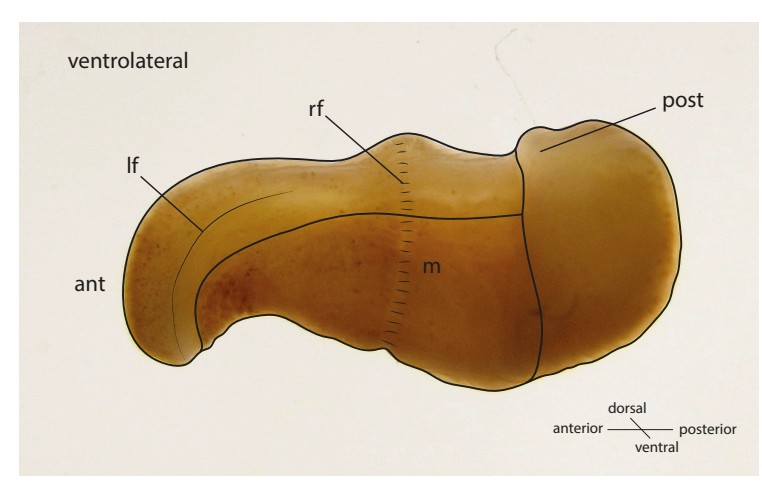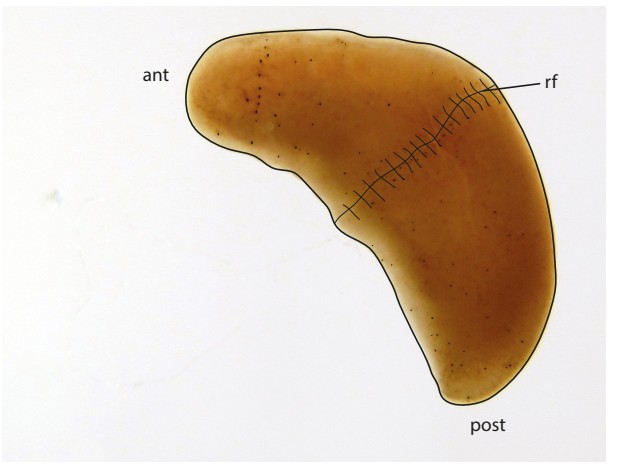

**Figure 1.** Schematic drawings of *X. bocki* showing the simple body organization of the marine vermiform animal. ant, anterior; post, posterior; lf, lateral furrow; rf, ring furrow; m, mouth opening.

(*Pryszcz and Gabaldón, 2016*), this increased to an N50 of ~62 kb over 23,094 contigs and scaffolds spanning ~121 Mb, and a longest scaffold of 960,978 kb (*Table 1*).

The final genome was obtained with Hi-C scaffolding using the program instaGRAAL (*Baudry et al., 2020*). The scaffolded genome has a span of 111 Mbp (117 Mbp including small fragments unincorporated into the HiC assembly) and an N50 of 2.7 Mbp (for contigs >500 bp). The assembly contains 18 megabase-scale scaffolds encompassing 72 Mbp (62%) of the genomic sequence, with 43% GC content. The original assembly indicated a repeat content of about 25% after a RepeatModeller-based RepeatMasker annotation ('Materials and methods'). As often seen in non-model organisms, about 2/3 of the repeats are not classified.

We used BRAKER (*Hoff et al., 2019*, *Hoff et al., 2016*) with extensive RNA-seq data, and additional single-cell UTR enriched transcriptome sequencing data to predict 15,154 gene models. A total of 9575 gene models (63%) are found on the 18 large scaffolds (which represent 62% of the total sequence). A total of 13,298 of our predicted genes (88%) have RNA-seq support. Although this proportion is at the low end of bilaterian gene counts, we note that our RNA-seq libraries were all taken from adult animals and thus may not represent the true complexity of the gene complement. We consider our predicted gene number to be a lower bound estimate for the true gene content.

The predicted *X. bocki* genes have a median coding length of 873 nt and a mean length of 1330 nt. Median exon length is 132 nt (mean 212 nt) and median intron length is 131 nt (mean 394 nt). Genes have a median of 4 exons and a mean of 8.5 exons. A total of 2532 genes have a single exon, of which 1381 are supported as having a single exon by RNA-seq (TPM > 1). A comparison of the exon, intron, and intergenic sequence content in *Xenoturbella* with those described in other animal genomes (*Francis and Wörheide, 2017*) shows that *X. bocki* falls within the range of other similarly sized metazoan genomes (*Figure 2*) for all these measures.

**Table 1.** Improvement of assembly and scaffolding metrics.

| Assembly step | # seqs | # reals | # Ns | Max length | N50 |
|---|---|---|---|---|---|
| Redundans contigs | 37,880 | 113,212,556 | 38,3327 | 206,709 | 8544 |
| Redundans scaffolds | 24,538 | 117,405,089 | 3,021,351 | 952,321 | 52,073 |
| Pre instaGRAAL | 23,094 | 117,396,873 | 3,534,582 | 960,978 | 61,989 |
| Final scaffolds | 27,939 | 107,712,917 | 3,328,069 | 8,757,424 | 2,730,651 |

Assessed with the jvci toolbox: https://github.com/tanghaibao/jcvi (*Tang, 2010*).

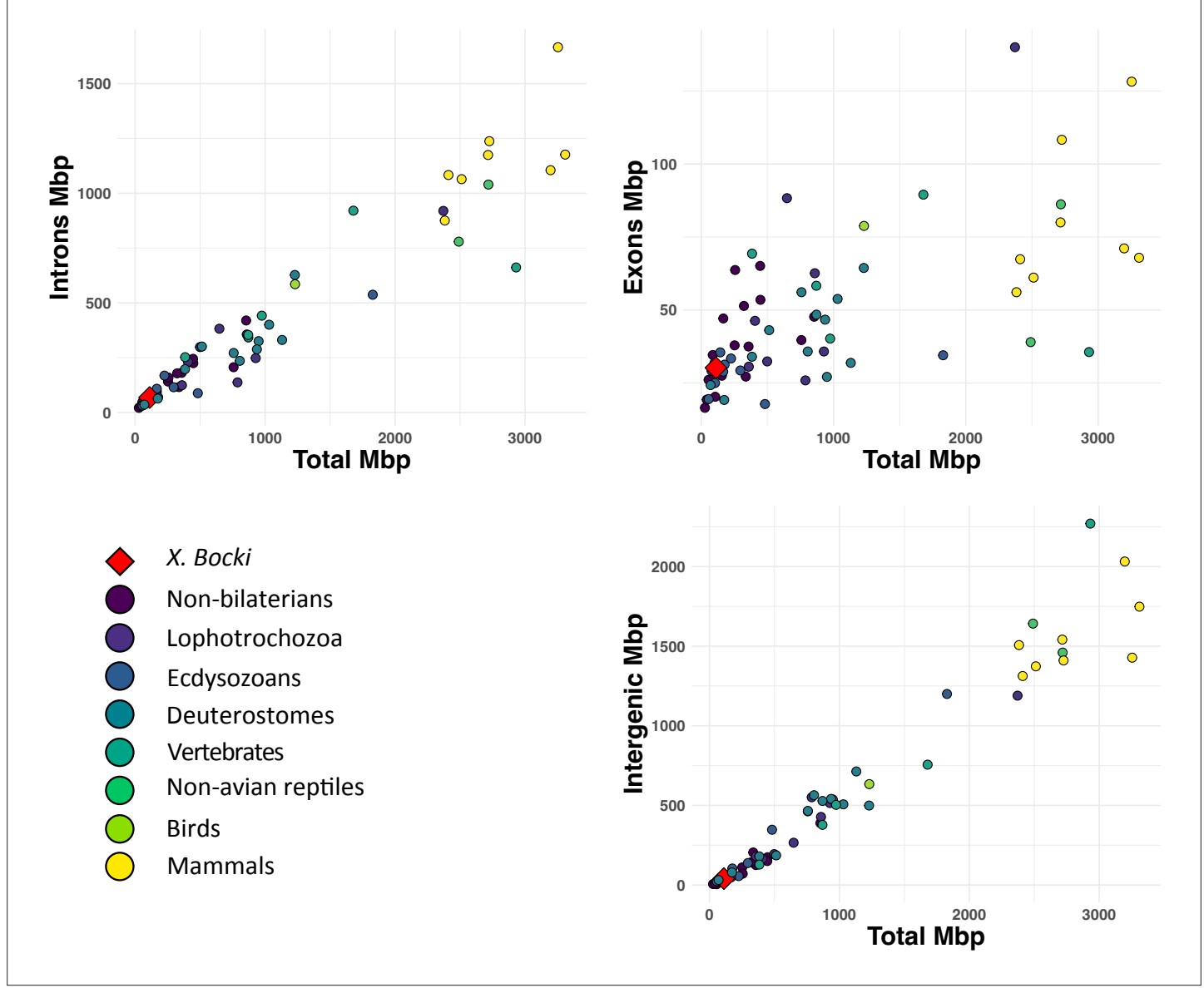

**Figure 2.** A comparison of total length of exons, intrans, and intergeneic space in the *X. bocki* genome with other metazoans (data from *Francis and Wörheide, 2017*). *X. bocki* does not appear to be an outlier in any of these comparisons.

## The genome of a co-sequenced *Chlamydia* species

We recovered the genome of a marine *Chlamydia* species from Illumina data obtained from one *X. bocki* specimen and from Oxford Nanopore data from a second specimen supporting previous microscopic analyses and single-gene PCRs suggesting that *X. bocki* is host to a species in the bacterial genus *Chlamydia*. The bacterial genome was found as five contigs spanning 1,906,303 bp (N50 of 1,237,287 bp), which were assembled into two large scaffolds. Using PROKKA (*Seemann, 2014*), we predicted 1738 genes in this bacterial genome, with 3 ribosomal RNAs, 35 transfer RNAs, and 1 transfer-messenger RNA. The genome is 97.5% complete for bacterial BUSCO (*Simão et al., 2015*) genes, missing only one of the 40 core genes.

Marine chlamydiae are not closely related to the group of human pathogens (*Dharamshi et al., 2020*), and we were not able to align the genome of the *Chlamydia*-related symbiont from *X. bocki* to the reference strain *Chlamydia trachomatis* F/SW4, nor to *Chlamydophila pneumoniae* TW-183. To investigate the phylogenetic position of the species co-occurring with *Xenoturbella*, we aligned the 16S rRNA gene from the *X. bocki*-hosted *Chlamydia* with orthologs from related species including

sequences of genes amplified from DNA/RNA extracted from deep-sea sediments. The *X. bocki*-hosted *Chlamydia* belong to a group designated as Simkaniaceae in *Dharamshi et al., 2020*, with the sister taxon in our phylogenetic tree being the *Chlamydia* species previously found in *X. westbladi* (*X. westbladi* is almost certainly a synonym of *X. bocki*) (*Rouse et al., 2016*; *Figure 3*).

To investigate whether the *X. bocki*-hosted *Chlamydia* might contribute to the metabolic pathways of its host, we compared the completeness of metabolic pathways in KEGG for the *X. bocki* genome alone and for the *X. bocki* genome in combination with the bacteria. We found only slightly higher completeness in a small number of pathways involved in carbohydrate metabolism, carbon fixation, and amino acid metabolism (see supplementary material), suggesting that the relationship is likely to be commensal or parasitic rather than a true symbiosis.

A second large fraction of bacterial reads, annotated as Gammaproteobacteria, were identified and filtered out during the data processing steps. These bacteria were also previously reported as potential symbionts of *X. bocki* (*Kjeldsen et al., 2010*). However, these sequences were not sufficiently well covered to reconstruct a genome, and we did not investigate them further.

## HGT into the *X. bocki* genome is low

Given the close association with bacteria, we were curious to see whether the *X. bocki* genome contains an elevated number of horizontally acquired genes. We did not find this to be the case. We were able to detect 56 potential horizontal gene transfer (HGT) events. Phylogenies generated using closest blast hits for each HGT candidate unveiled one of the 56 genes to be of chlamydial origin and thus likely originating from a bacterial contig. A number of HGT candidates appear to be of proteobacterial origin, coding for a functionally diverse set of proteins. In summary, 0.35% of the *X. bocki* genes we have identified might be horizontally acquired. See supplementary online material for alignments and gene trees.

## A phylogenetic gene presence/absence matrix supports Xenambulacraria

The general completeness of the *X. bocki* gene set allowed us to use the presence and absence of genes identified in our genomes as a source of information to find the best supported phylogenetic position of the Xenacoelomorpha. We conducted two separate phylogenetic analyses of gene presence/absence data: one including the fast-evolving Acoelomorpha and one without. In both analyses, the best tree grouped *Xenoturbella* with the Ambulacraria (*Figure 4a*). The analysis including acoels, however, placed the acoels as the sister group to Nephrozoa separate from *Xenoturbella* (*Figure 4b*). There are two explanations for this finding. The first would be that the Xenacoelomorphs are paraphyletic; that *Xenoturbella* is the sister group of the Ambulacraria and Acoelomorpha the sister group of Nephrozoa. Because many other studies have shown the monophyly of Xenacoelomorpha to be robust (*Philippe et al., 2019*; *Cannon et al., 2016*; *Rouse et al., 2016*; *Srivastava et al., 2014*; *Philippe et al., 2011*; *Bourlat et al., 2006*; *Ueki et al., 2019*), we do not think this a plausible explanation. The second explanation of this observation is that it is the result of systematic error caused by a high rate of gene loss or by orthologs being incorrectly scored as missing due to higher rates of sequence evolution in acoelomorphs (*Natsidis et al., 2021*). Under this second scenario, we consider it more likely that, of the two clades, it is the Acoelomorpha not *Xenoturbella* that are wrongly placed and that the position of *Xenoturbella* represents the more likely position of the entire phylum of Xenacoelomorpha. We note that under both scenarios the focus of our work, *Xenoturbella*, is the sister group of the Ambulacraria though the implied error suggests that using gene presence/absence may not be the ideal way to solve difficult phylogenetic problems.

## The *X. bocki* molecular toolkit is typical of bilaterians

One of our principal aims was to ask whether the *Xenoturbella* genome lacks characteristics otherwise present in the Bilateria. We found that for the Metazoa gene set in BUSCO (v5) the *X. bocki* proteome translated from our gene predictions is 82.5% complete and ~90% complete when partial hits are included (82 and 93%, respectively, for the Eukaryote gene set). This estimate is even higher in the acoel *H. miamia*, which was originally reported to be 90% (*Gehrke et al., 2019*), but in our re-analysis was 95.71%. In comparison, the morphologically highly simplified and fast-evolving annelid *Intoshia linei* (*Schiffer et al., 2018*) has a genome of fewer than 10,000 genes (*Mikhailov et al., 2016*) and in

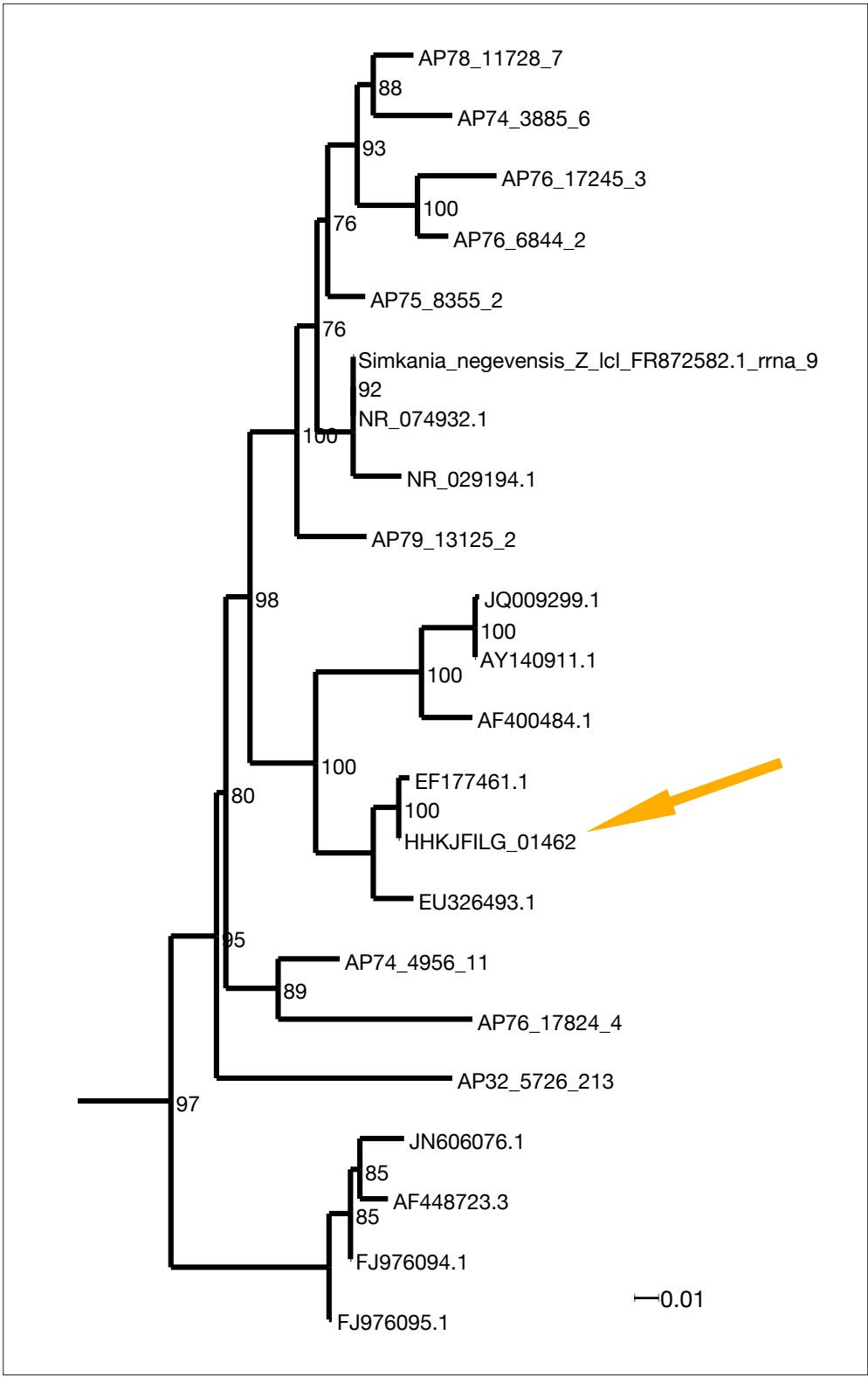

**Figure 3.** *X. bocki* harbors a marine *Chlamydiae* species as potential symbiont. In the phylogenetic analysis of 16S rDNA (ML: GTR + F + R7; bootstrap values included) the bacteria in our *X. bocki* isolate (arrow) are sister lo a previous isolate from *X. westbladi. X. westbladi* is most likely a mis-identification of *X. bocki.*

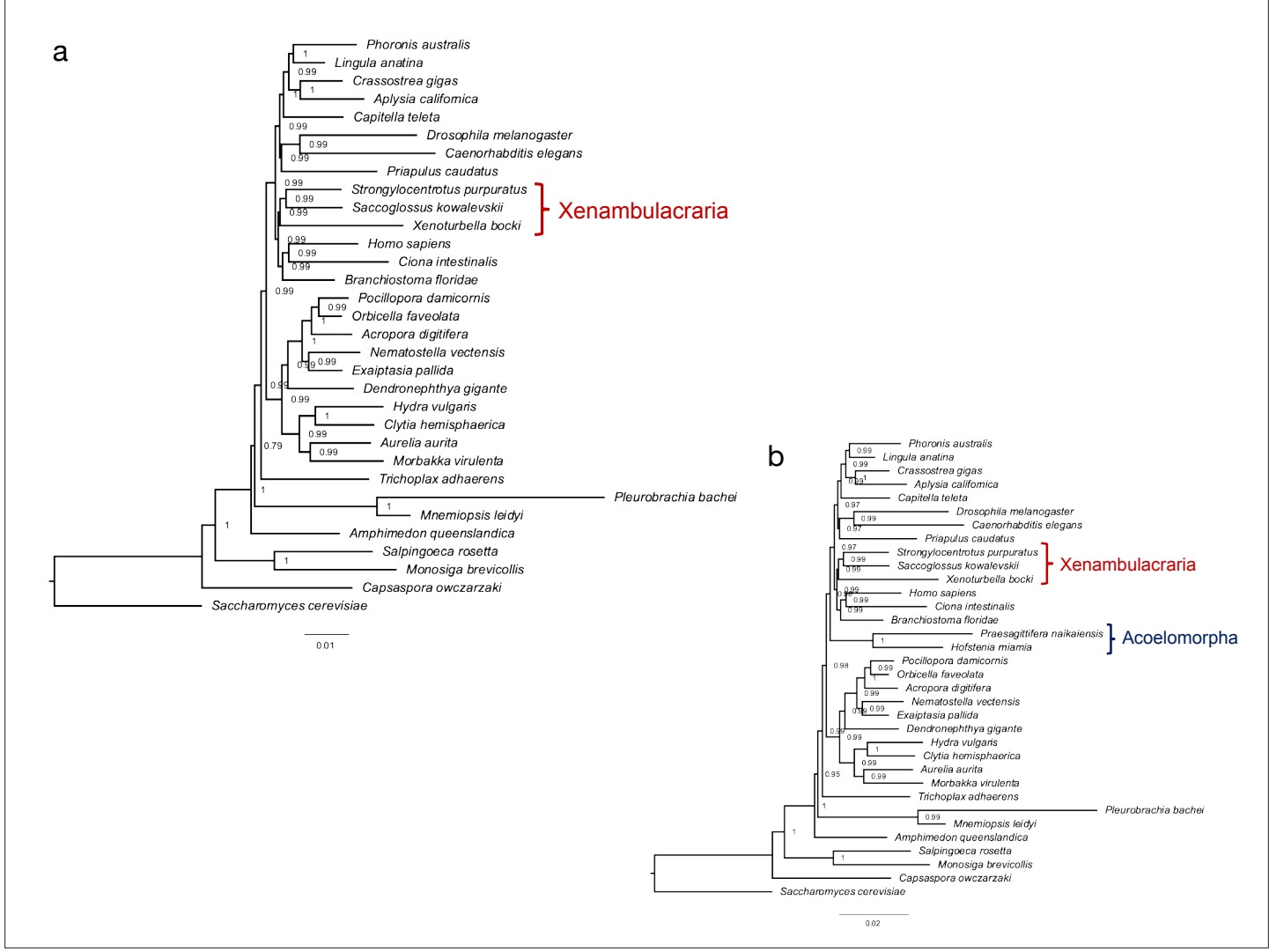

**Figure 4.** A phylogeny based on the presence and absence of genes calculated with OMA. Both analysis (**a**) and (**b**) confirm Xenambulacraria, that is, Xenoturbellida in a group with Echinoderms and Hemichordates. Inclusion of the acoel flatworms places these as sister to all other Bilateria (**b**). This placement appears as an artifact due to the very fast evolution in this taxon, in particular as good evidence exists for uniting Xenoturbellida and Acoela (**Philippe et al., 2019**; **Cannon et al., 2016**; **Rouse et al., 2016**; **Srivastava et al., 2014**; **Philippe et al., 2011**; **Bourlat et al., 2006**; **Ueki et al., 2019**).

our analysis is only ~64% complete for the BUSCO (v5) Metazoa set. The model nematode *Caenorhabditis elegans* is ~79% complete for the same set. Despite the morphological simplicity of both *Xenoturbella* and *Hofstenia*, these Xenacoelomorpha are missing few core genes compared to other bilaterian lineages that we perceive to have undergone a high degree of morphological evolutionary change (such as the evolution of miniaturization, parasitism, sessility, etc.).

Using our phylogenomic matrix of gene presence/absence (see above), we identified all orthologs that could be detected both in Bilateria (in any bilaterian lineage) and in any non-bilaterian; ignoring HGT and other rare events, these genes must have existed in Urbilateria (and, of less interest to us, in Urmetazoa). The absence of any of these bilaterian genes in any lineage of Bilateria must therefore be explained by loss of the gene. All individual bilaterian genomes were missing many of these orthologs but Xenacoelomorphs and some other bilaterians lacked more of these than did other taxa. The average numbers of these genes present in bilaterians = 7577; *Xenoturbella* = 5459; *Hofstenia* = 5438; *Praesagittifera* = 4280; *Drosophila* = 4844 and *Caenorhabditis* = 4323.

To better profile the *Xenoturbella* and xenacoelomorph molecular toolkits, we used OrthoFinder to conduct orthology searches in a comparison of 155 metazoan and outgroup species, including the

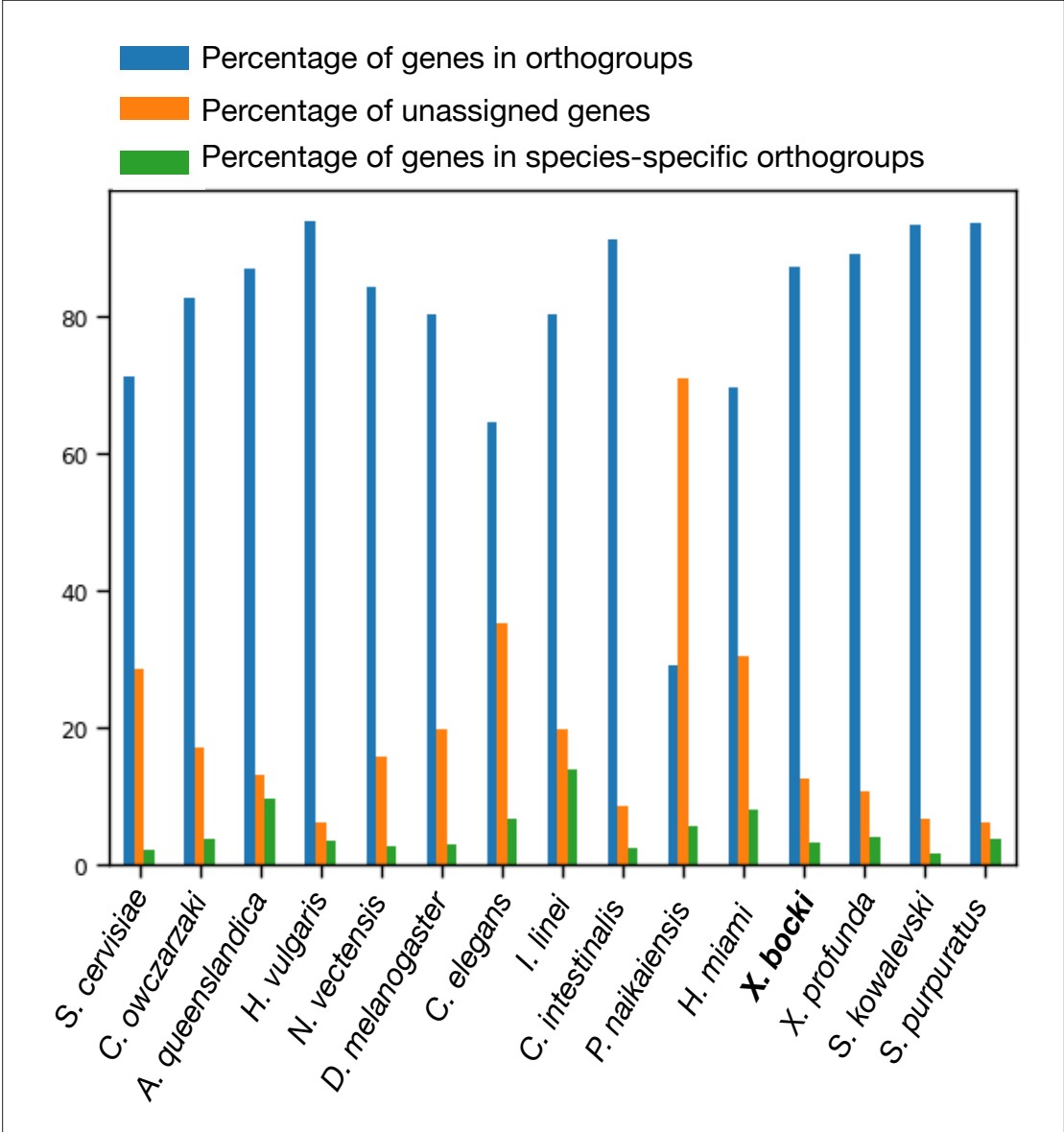

**Figure 5.** In our orthology screen, *X. bocki* shows similar percentages of genes in orthogroups, unassigned genes, and species-specific orthogroups as other well-annotated enomes.

transcriptomes of the sister species *Xenoturbella profunda* and a draft genome of the acoel *Paratomella rubra* we had available, as well as the *Hofstenia* and *Praesagittifera* proteomes (***Supplementary file 1***). For each species, we counted, in each of the three Xenacoelomorphs, the number of orthogroups for which a gene was present. The proportion of orthogroups containing an *X. bocki* and *X. profunda* protein (87.4 and 89.2%) are broadly similar to the proportions seen in other well-characterized genomes, for example, *S. purpuratus* proteins (93.8%) or *N. vectensis* proteins (84.3%) (***Figure 5***). In this analysis, the fast-evolving nematode *C. elegans* appears as an outlier, with only ~64% of its proteins in orthogroups and ~35% unassigned. Both *Xenoturbella* species have an intermediate number of unassigned genes of ~11–12%. Similarly, the proportion of species-specific genes (~14% of all genes) corresponds closely to what is seen in most other species (with the exception of the parasitic annelid *I. linei*, ***Figure 5***).

## Idiosyncrasies of *Xenoturbella*

In order to identify sets of orthologs specific to the two *Xenoturbella* species, we used the kinfin software (***Laetsch and Blaxter, 2017a***) and found 867 such groups in the OrthoFinder clustering. We

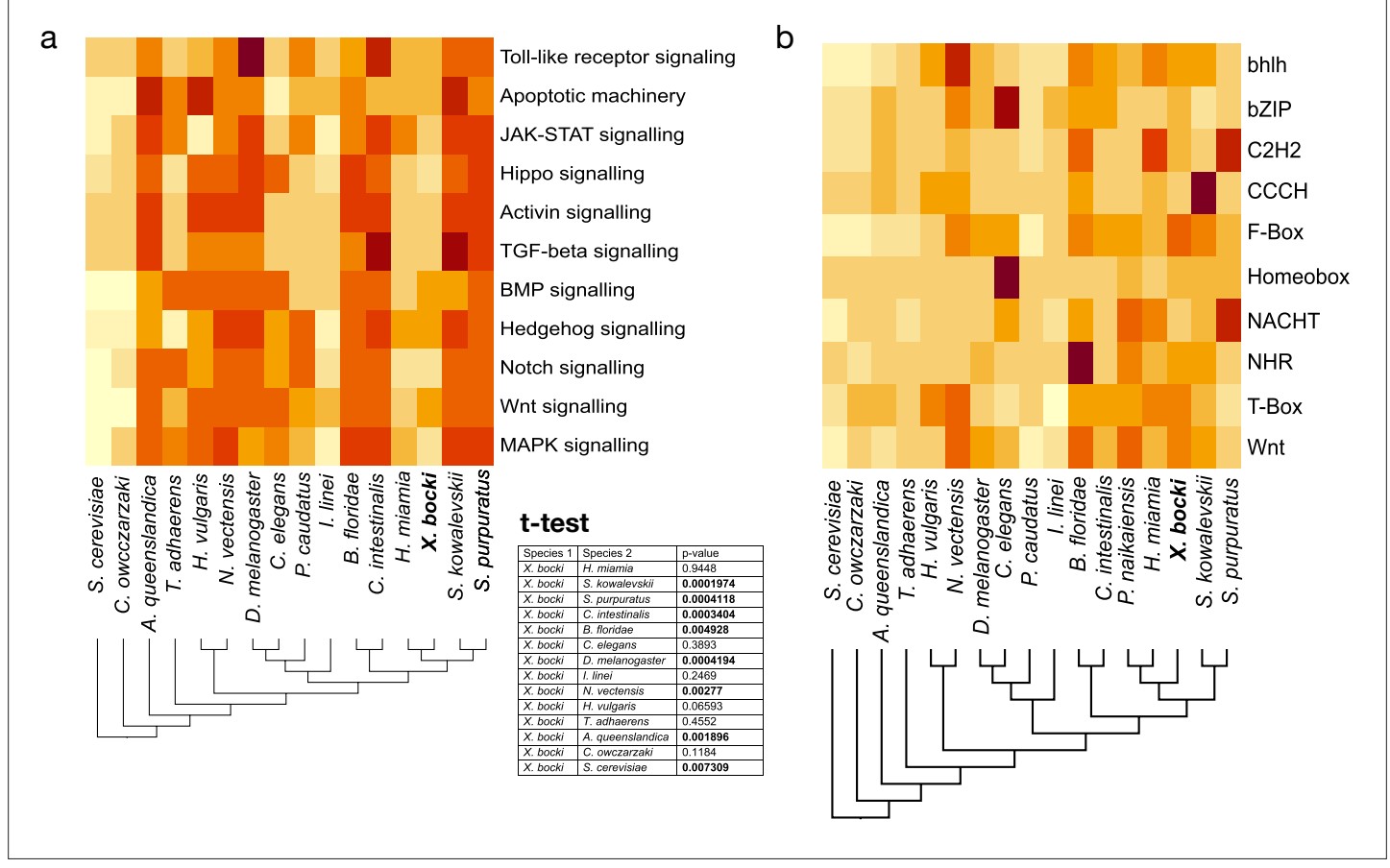

**Figure 6.** The heatmaps show a comparative measure of relative completeness of signaling pathways based on KEGG and assessed with GenomeMaple or abundance of genes in a given gene-family based on InterProsScan annotations. (**a**) Cell signaling pathways in *X. bocki* are functionally complete, but in comparison to other species contain less genes. The overall completeness is not significantly different to, for example, the nematode *C. elegans* (inset, *t*-test). (**b**) The number of family members per species in major gene families (based on Pfam domains), like transcription factors, fluctuates in evolution. The *X. bock*i genome does not appear to contain particularly less or more genes in any of the analyzed families. Due to the comparative nature of the assay, no 'true' scale *can* be given: darker colors indicate higher comparative completeness. Schematic cladograms are drawn by the authors.

profiled these genes based on Pfam domains and GO terms derived from InterProScan. While these *Xenoturbella* specific proteins fall into diverse classes, we did see a considerable number of C-type lectin, Immunoglobulin-like, PAN, and Kringle domain containing Pfam annotations. Along with the cysteine-rich secretory protein family and the G-protein-coupled receptor activity GO terms, these genes and families of genes may be interesting for future studies into the biology of *Xenoturbella* in its native environment.

## Gene families and signaling pathways are retained in *X. bocki*

In our orthology clustering, we did not see an inflation of *Xenoturbella*-specific groups in comparison to other taxa, but also no conspicuous absence of major gene families (**Figure 6**). Family numbers of transcription factors like Zinc-fingers or homeobox-containing genes, as well as, for example, NACHT-domain encoding genes seem to be neither drastically inflated nor contracted in comparison to other species in our InterProScan-based analysis.

To catalogue the completeness of cell signaling pathways, we screened the *X. bocki* proteome against KEGG pathway maps using GenomeMaple (**Takami et al., 2016**). The *X. bocki* gene set is largely complete in regard to the core proteins of these pathways, while an array of effector proteins is absent (**Figure 6**). In comparison to other metazoan species, as well as to a unicellular choanoflagellate and a yeast, the *X. bocki* molecular toolkit has significantly lower KEGG completeness than

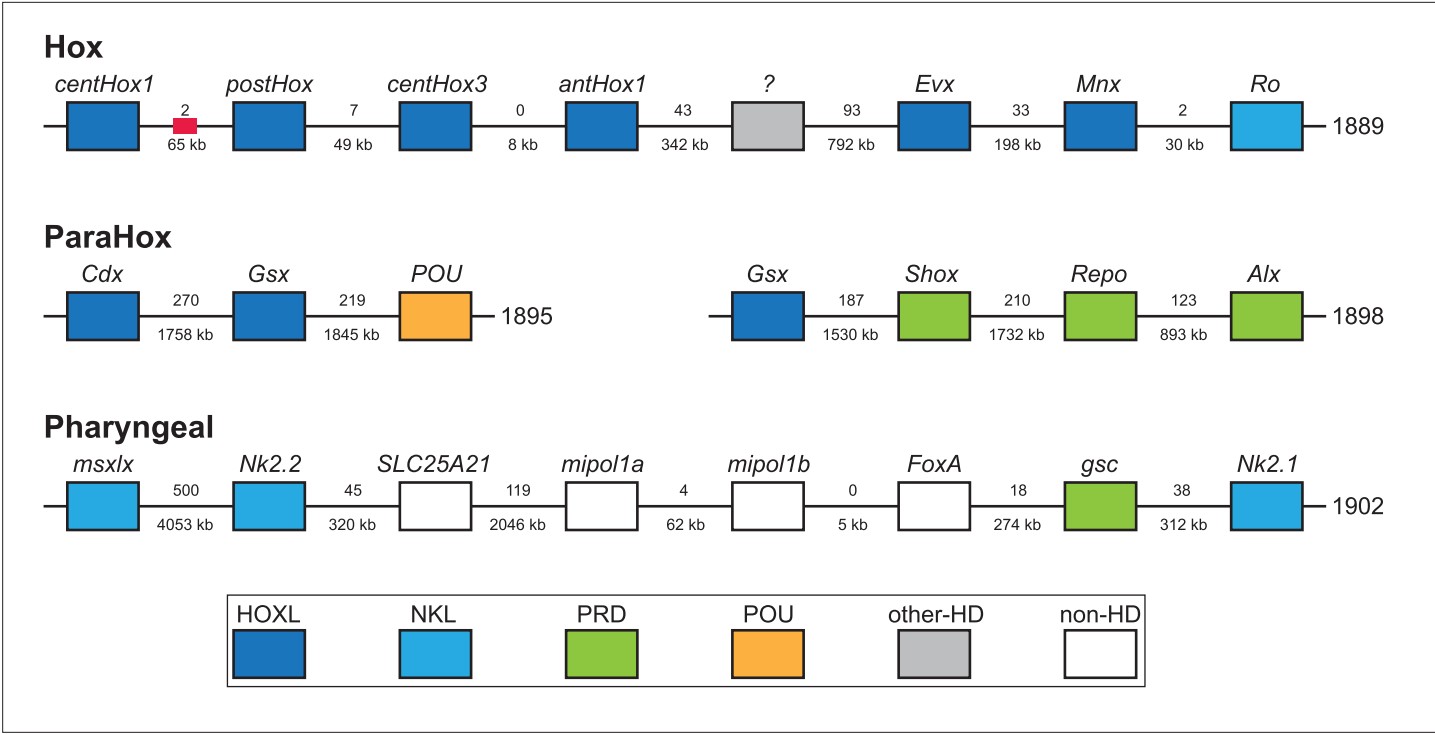

**Figure 7.** *X. bocki* has five HOX genes, which are located in relatively close proximity on one of our chromosome-size scaffolds. Similar clusters exist for the ParaHox and 'pharyngeal' genes. Numbers between genes are distance (below) and number of genes between (below). Colors indicate gene families. Red box marks the position of a partial Hox gene. The '?' gene has an unresolved homeodomain identity.

morphologically complex animals such as the sea urchin and amphioxus (*t*-test; *Figure 6*). *Xenoturbella* is, however, not significantly less complete compared to other bilaterians considered to have low morphological complexity and which have been shown to have reduced gene content, such as *C. elegans*, the annelid parasite *I. linei*, or the acoel *H. miamia* (*Figure 6*).

### Clustered homeobox genes in the *X. bocki* genome

Acoelomorph flatworms possess three unlinked HOX genes, orthologs of anterior (Hox1), central (Hox4/5 or Hox5), and posterior Hox (HoxP). In contrast, previous analysis of *X. bocki* transcriptomes identified one anterior, three central, and one posterior Hox genes. We identified clear evidence of a syntenic Hox cluster with four Hox genes (centHox1, postHox, centHox3, and antHox1) in the *X. bocki* genome (*Figure 7*). There was also evidence of a fragmented annotation of centHox2, split between the four gene Hox cluster and a separate scaffold (*Figure 7*). In summary, this suggests that all five Hox genes form a Hox cluster in the *X. bocki* genome, but that there are possible unresolved assembly errors disrupting the current annotation. We also identified other homeobox genes on the Hox cluster scaffold, including Evx (*Figure 7*).

Along with the Hox genes, we surveyed other homeobox genes that are typically clustered in Bilateria. The canonical bilaterian ParaHox cluster contains three genes Cdx, Xlox (=Pdx), and Gsx. We identified Cdx and a new Gsx annotation on the same scaffold, as well as a previously reported Gsx paralog on a separate scaffold. This indicates partial retention of the ParaHox cluster in *X. bocki* along with a duplication of Gsx. On both of these ParaHox-containing scaffolds, we observed other homeobox genes.

Hemichordates and chordates have a conserved cluster of genes involved in patterning their pharyngeal pores – the so-called 'pharyngeal cluster'. The homeobox genes of this cluster (Msxlx, Nk2-1/2/4/8) were present on a single *X. bocki* scaffold. Another pharyngeal cluster transcription factor, the Forkhead containing Foxa, and 'bystander' genes from that cluster including Egln, Mipol1, and Slc25a21 are found in the same genomic region. Different subparts of the cluster are found in non-bilaterians and protostomes, and the cluster may well be plesiomorphic for the Bilateria rather than a deuterostome synapomorphy (*Kapli et al., 2021*).

## The *X. bocki* neuropeptide complement is larger than previously thought

A catalog of acoelomorph neuropeptides was previously described using transcriptome data (*Thiel et al., 2018a*). We have discovered 12 additional neuropeptide genes and 39 new neuropeptide receptors in *X. bocki* adding 6 bilaterian peptidergic systems to the *Xenoturbella* catalog (NPY-F; MCH/Asta-C; TRH; ETH; CCHa/Nmn-B; Np-S/CCAP), and 6 additional bilaterian systems to the Xenacoelomorpha catalog (Corazonin; Kiss/GPR54; GPR83; 7B2; Trunk/PTTH; NUCB2), making a total of 31 peptidergic systems (*Figure 8*).

Among the ligand genes, we identified six new repeat-containing sequences. One of these, the LRIGamide-peptide, had been identified in Nemertodermatida and Acoela and its loss in *Xenoturbella* had been proposed (*Thiel et al., 2018a*). We also identified the first 7B2 neuropeptide and NucB2/Nesfatin genes in Xenacoelomorpha. Finally, we identified three new *X. bocki* insulin-like peptides, one of them sharing sequence similarity and an atypical cysteine pattern with the Ambulacrarian octinsulin, constituting a potential synapomorphy of Xenambulacraria (see https://doi.org/10.5281/zenodo.6962271).

Our searches also revealed the presence of components of the arthropod moulting pathway components (PTTH/trunk, NP-S/CCAP, and Bursicon receptors), which have recently been shown to be of ancient origin (*de Oliveira et al., 2019*). We further identified multiple paralogs of the Tachykinin, Rya/Luqin, tFMRFa, Corazonin, Achatin, CCK, and Prokineticin receptor families. Two complete *X. bocki* Prokineticin ligands were also found in our survey (*Figure 8*).

Chordate Prokineticin ligands possess a conserved N-terminal 'AVIT' sequence required for the receptor activation (*Negri and Ferrara, 2018*). This sequence is absent in arthropod Astakine, which instead possess two signature sequences within their Prokineticin domain (*Ericsson and Söderhäll, 2018*). To investigate Prokineticin ligands in Xenacoelomorpha, we compared the sequences of their Prokineticin ligands with those of other bilaterians (*Figure 8*). Our alignment reveals clade-specific signatures already reported in Ecdysozoa and Chordata sequences, but also two new signatures specific to Lophotrochozoa and Cnidaria sequences, as well as a very specific 'K/R-RFP-K/R' signature shared only by ambulacrarian and *X. bocki* sequences. The shared Ambulacrarian/Xenacoelomorpha signature is found at the same position as the Chordate sequence involved in receptor activation – adjacent to the N-terminus of the Prokineticin domain (*Figure 8*).

## The *X. bocki* genome contains most of the bilaterian miRNAs reported missing from acoels

microRNAs have previously been used to investigate the phylogenetic position of the acoels and *Xenoturbella*. The acoel *S. roscoffensis* lacks some protostome and bilaterian miRNAs, and this lack was interpreted as supporting the position of acoels as sister group to the Nephrozoa. Based on shallow 454 microRNA sequencing (and sparse genomic traces) of *Xenoturbella*, some of the bilaterian miRNAs missing from acoels were found – 16 of the 32 expected metazoan (1 miRNA) and bilaterian (31 miRNAs) microRNA families – of which six could be identified in genome traces (*Philippe et al., 2011*).

By deep sequencing two independent small RNA samples, we have now identified the majority of the missing metazoan and bilaterian microRNAs and identified them in the genome assembly (*Figure 9*). Altogether, we found 23 out of 31 bilaterian microRNA families (35 genes including duplicates); the single known Metazoan microRNA family (MIR-10) in two copies; the Deuterostome-specific MIR-103; and 7 *Xenoturbella*-specific microRNAs, giving a total of 46 microRNA genes. None of the protostome-specific miRNAs were found. We could not confirm in the RNA sequences or new assembly a previously identified, and supposedly xenambulacrarian-specific, MIR-2012 ortholog.

## The *X. bocki* genome retains ancestral metazoan linkage groups

The availability of chromosome-scale genomes has made it possible to reconstruct 24 ancestral linkage units broadly preserved in bilaterians (*Simakov et al., 2020*). In fast-evolving genomes, such as those of nematodes, tunicates, or platyhelminths, these ancestral linkage groups (ALGs) are often dispersed

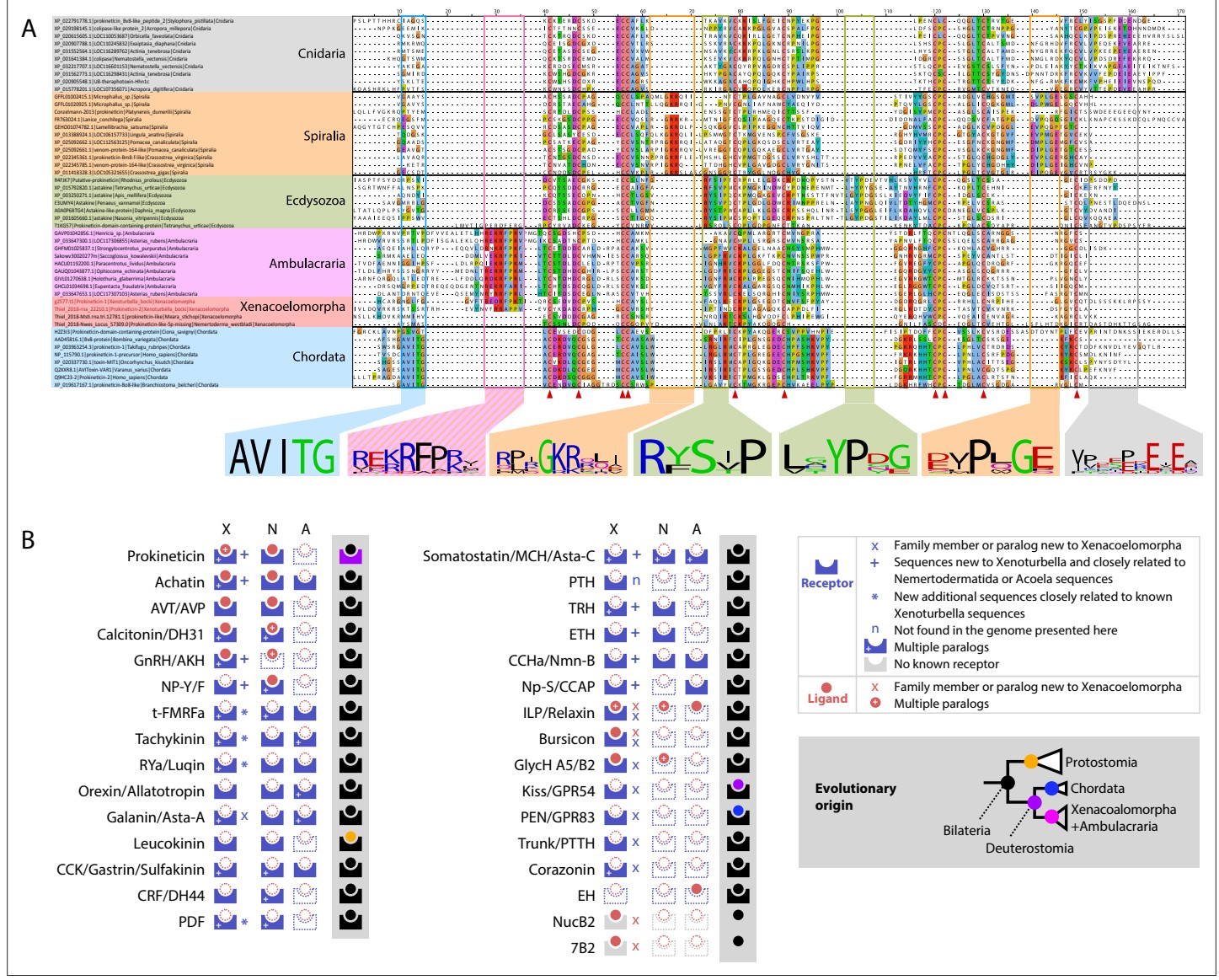

**Figure 8.** *X.bocki* genome contains genes for most bilaterian specific peptidergic system and a prokineticin gene containing a signature sequence shared with ambulacraria. (**a**) Sequence alignment of Cnidarian Colipase-like protein, Ecdysozoan Astakine-like protein and Spiralian, Chordates and Xenacoelomorpha Prokineticin-like proteins show conserved cysteine positions (highlighted by red triangle), as well as clade specific signature sequences sequences among which a "K/R-RFP-K/R" sequence shared only by ambulacrarians and *X. bocki*. The signature previously reported for Ecdysozoa and chordata, as well as new signatures we found in Spiralia and Cnidaria is absent from ambulacrarians and *X. bocki* prokineticin ligand sequences. Sequences are available as *Figure 8—source data 1*; alignment files are available at https://doi.org/10.5281/zenodo.6962271. (**b**) Peptidergic systems found in Xenoturbella (X), Nemertodermatida (N) and Acoelomorpha (A). Novel findings are highlighted in the top right inset. Color of schemes and inset cladogram nodes on grey background depicts the evolutionary origin of peptidergic systems in accordance with our analysis: bilaterian, protostomian, chordate, xenacoelomorph + ambulacrarian last common ancestors respectively. 7B2, Neuroendocrine protein 7B2; AKH, adipokinetic hormone; Asta-A, Allatostatin-A; Asta-C, Allatostatin-C; AVP, arginine vasopressin; AVT, Arginine vasotocin; CCAP, crustacean cardioactive peptide; CCHa, CCHamide peptide; CCK, cholecystokinin; CRF, Corticotropin-releasing factor; DH31, diuretic hormone 31; DH44, diuretic hormone 44; EH, eclosion hormone; GlycH A5, Glycoprotein Hormone alpha5; GlycH B2, Glycoprotein Hormone beta2; GnRH, Gonadotropin Releasing Hormone; GPR54, G Protein-Coupled Receptor 54; GPR83, G Protein-Coupled Receptor 83; ILP, Insulin-like peptide; Kiss, Kisspeptine; MCH, melanin concentrating hormone; Nmn-B, Neuromedin B; Np-S, Neuropeptide S; NP-Y/F, Neuropeptide Y/F; NucB2, nucleobindin 2; PDF, Pigment-dispersing factor; PEN, neuroendocrine peptide PEN; PTTH, Prothoracicotropic hormone; RYa, RYamide peptide; t-FMRFa, trochozoan-FMRFamide peptide.

The online version of this article includes the following source data and figure supplement(s) for figure 8:

**Source data 1.** *Xenoturbella bocki* neuropeptide sequences.

**Source data 2.** *Xenoturbella bocki* neuropeptide receptor sequences.

*Figure 8 continued on next page*

*Figure 8 continued*

**Figure supplement 1.** Radial tree representation of the phylogenetic analysis of bilaterian glycoprotein hormone and Bursicon.

**Figure supplement 2.** Radial tree representation of the sequence similarities analysis of bilaterian insulin-related peptides.

**Figure supplement 3.** Circular tree representation of the phylogenetic analysis of bilaterian Leucine-rich repeat-containing G-protein coupled Receptors (Rhodopsin type G-protein coupled Receptors delta).

**Figure supplement 4.** Circular tree representation of the phylogenetic analysis of bilaterian Rhodopsin type G-protein coupled Receptors beta and gamma.

**Figure supplement 5.** Circular tree representation of the phylogenetic analysis of bilaterian Tyrosine kinase Receptors.

**Figure supplement 6.** Circular tree representation of the phylogenetic analysis of bilaterian Secretin type G-protein coupled Receptors.

and/or extensively fused (supplementary material). We were interested in testing whether the general conservation of the gene content in *X. bocki* is reflected in its genome structure.

We compared the genome of *Xenoturbella* to several other metazoan genomes and found that it has retained most of these ancestral bilaterian units: 12 chromosomes in the *X. bocki* genome derive from a single ALG, 5 chromosomes are made of the fusion of 2 ALGs, and 1 *Xenoturbella* chromosome is a fusion of 3 ALGs, as highlighted with the comparison of ortholog content with amphioxus, the sea urchin, and the sea scallop (*Figure 10* and supplementary material).

One ALG that has been lost in chordates but not in ambulacrarians nor in mollusks (ALG R in sea urchin and sea scallop) is detectable in *X. bocki* (*Figure 10*), while *X. bocki* does not show the fusions that are characteristic of lophotrochozoans.

We also attempted to detect some pre-bilaterian arrangement of ancestral linkage: for instance, *Simakov et al., 2022* predicted that several pre-bilaterian linkage groups successively fused in the bilaterian lineage to give ALGs A1, Q, and E. These ALGs are all represented as single units in *X. bocki* in common with other Bilateria. None of the inferred pre-bilaterian chromosomal arrangements that could have provided support for the Nephrozoa hypothesis were found *in X. bocki,* although of course this does not rule out Nephrozoa.

## One *X. bocki* chromosomal fragment appears aberrant

The smallest of the 18 large scaffolds in the *X. bocki* genome did not show strong 1:1 clustering with any scaffold/chromosome of the bilaterian species we compared it to. To exclude potential contamination in the assembly as a source for this contig, we examined the orthogroups to which the genes from this scaffold belong. We found that *X. profunda* (*Rouse et al., 2016*), for which a transcriptome is available, was the species that most often occurred in the same orthogroup with genes from this scaffold (41 shared orthogroups), suggesting the scaffold is not a contaminant.

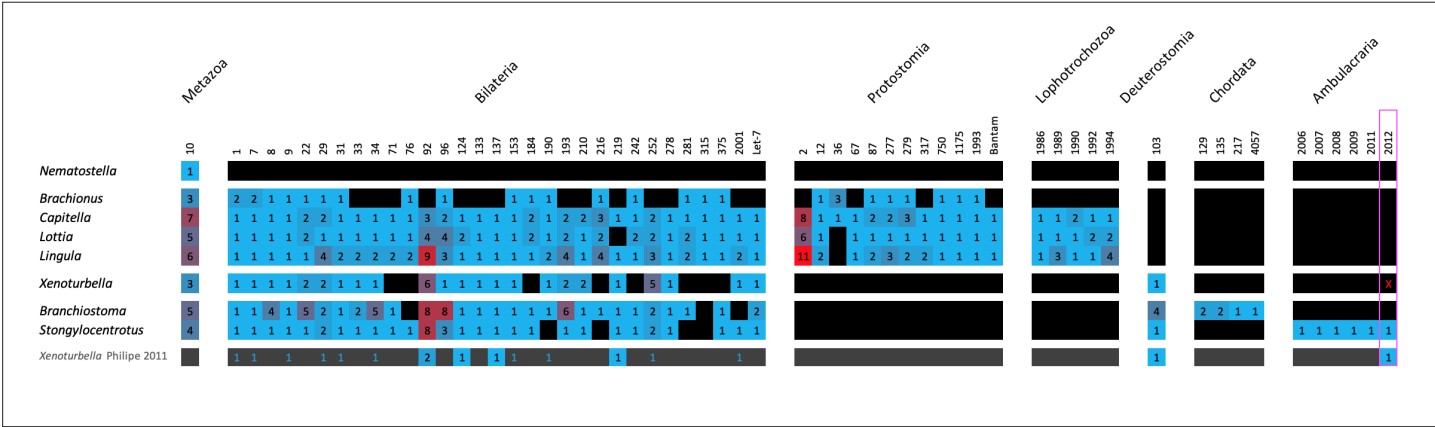

**Figure 9.** The rev sed microRNA complement of *X. bocki* has a near-complete set of metazoan, bilaterian, and deuterostome families and genes. Presence (color) and absence (black) of microRNA families (column), paralog numbers (values and heatmap coloring) organized in node-specific blocks in a range of representative protostome and deuterostome species compared with *Xenoturbella* (species from MirGeneDB 2.1; *Fromm et al., 2022*). The bottom row depicts 2011 complement by *Philippe et al., 2011* (blue numbers on black depict detected miRNA reads, but lack of genomic evidence). Red 'x' in the pink box highlights the lack of evidence for an Ambulacraria-specific microRNA in *X. bocki*.

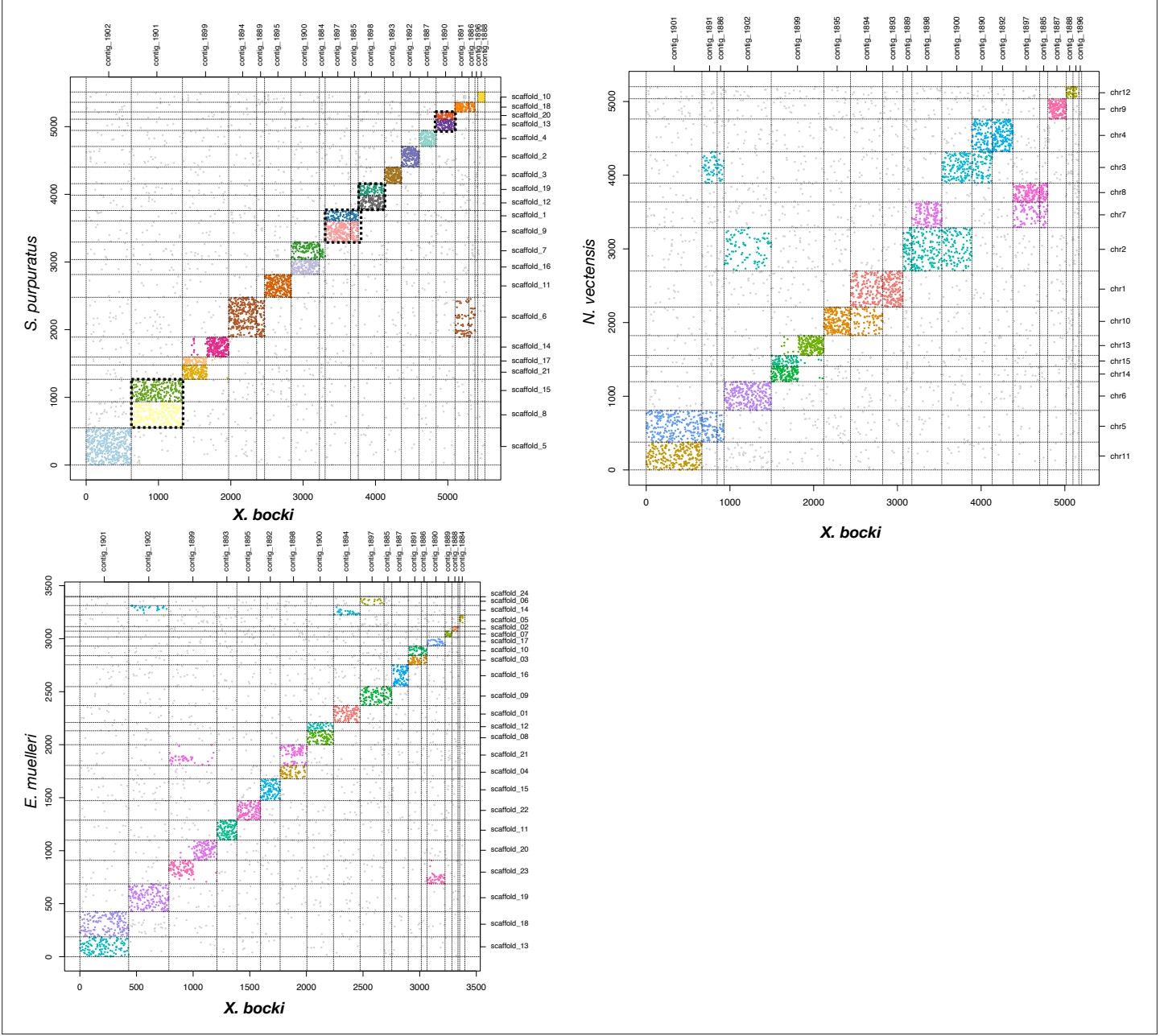

**Figure 10.** A comparison of scaffolds in the *X. bocki* genome with other Metazoa. 17 of the 18 large scaffolds in the *X. bocki* genome are linked via synteny to distinct chromosomal scaffolds in these species.

The online version of this article includes the following figure supplement(s) for figure 10:

**Figure supplement 1.** Conservation of metazoan synteny and methylation in *X. bocki*.

**Figure supplement 2.** Intergenomic comparison of *X. bocki* and *E. muelleri* highlighting synteny connections between the aberrant scaffold c1896 and scaffolds across the sponge genome.

We did observe links between the aberrant scaffold and several scaffolds from the genome of the sponge *Ephydatia muelleri* , but could not detect distinct synteny relationships to a single scaffold in other species. In line with this, genes on the scaffold show a different age structure compared to other scaffolds, with both more older genes (pre bilaterian) and more *Xenoturbella*-specific genes (*Figure 11*; supported by Ks statistics, supplementary material). This aberrant scaffold also had significantly lower levels of methylation than the rest of the genome.

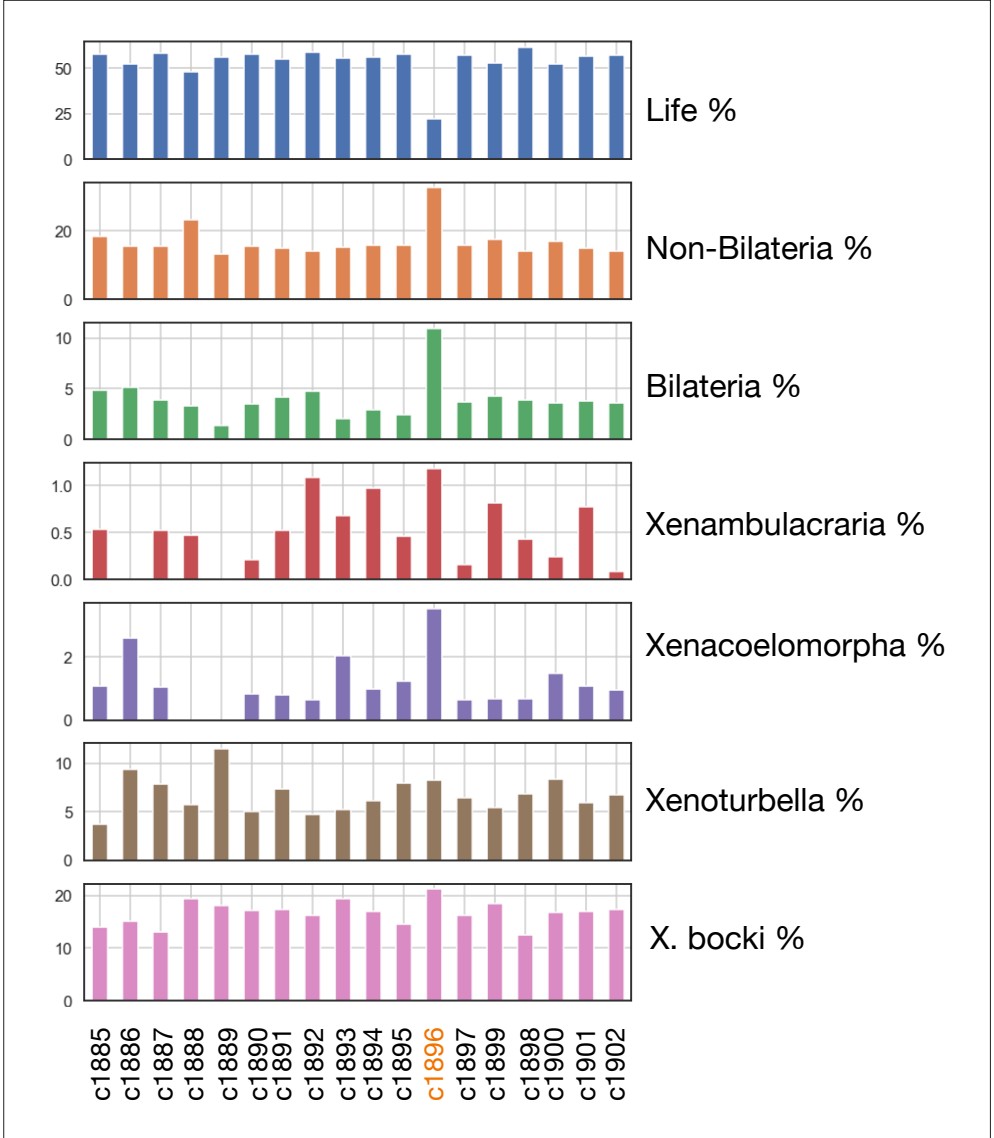

**Figure 11.** Phylostratigraphic age distribution of genes on all major scaffolds in the *X. bocki* genome. One scaffold (c1896), which showed no synteny to a distinct chromosomal scaffold in the other metazoan species, also had a divergent gene age structure in comparison to other *X. bocki* scaffolds.

## Discussion

The phylogenetic positions of *Xenoturbella* and the Acoelomorpha have been controversial since the first molecular data from these species appeared over 25 years ago. Today we understand that they constitute a monophyletic group of morphologically simple worms (*Telford, 2008*; *Philippe et al., 2011*; *Hejnol, 2015*), but there remains a disagreement over whether they represent a secondarily simplified sister group of the Ambulacraria or a primitively simple sister group to all other Bilateria. Here we wanted to analyze the genome of *X. bocki* to glean insights into their biology from a new perspective.

Previous analyses of the content of genomes, especially of Acoela, have found a small number of Hox genes and of microRNAs of acoels, and this has been interpreted as representing an intermediate stage on the path to the ~8 Hox genes and 30 odd microRNAs of the Nephrozoa. A strong version of the Nephrozoa idea would go further than these examples and anticipate, for example, a genome-wide paucity of bilaterian genes, GRNs, and biochemical pathways and/or an arrangement of chromosomal segments intermediate between those of the Eumetazoa and the Nephrozoa.

One criticism of the results from analyses of acoel genomes is that the Acoelomorpha have evolved rapidly (their long branches in phylogenetic trees showing high rates of sequence change). This rapid evolution might plausibly be expected to correlate with other aspects of rapid genome evolution such as higher rates of gene loss and chromosomal rearrangements, leading to significant differences from other Bilateria. The more normal rates of sequence evolution observed in *Xenoturbella* therefore recommend it as a more appropriate xenacoelomorph to study with fewer apomorphic characters expected.

We have sequenced, assembled, and analyzed a draft genome of *X. bocki*. To help with annotation of the genome, we have also sequenced miRNAs and small RNAs as well as using bisulfite sequencing, Hi-C, and Oxford Nanopore Technologies sequencing. We compared the gene content of the *Xenoturbella* genome to species across the Metazoa and its genome structure to several other high-quality draft animal genomes.

We found the *X. bocki* genome to be fairly compact, but not unusually reduced in size compared to many other bilaterians. It appears to contain a similar number of genes (~15,000) as other animals, for example, from the model organisms *Drosophila melanogaster* (>14,000) and *C. elegans* (~20,000). The BUSCO completeness, as well as a high level of representation of *X. bocki* proteins in the orthogroups of our 155 species orthology screen, indicates that we have annotated a near-complete gene set. Surprisingly, there are fewer genes than in the acoel *Hofstenia* (>22,000; BUSCO_v5 score ~95%). This said, of the genes found in Urbilateria (orthogroups in our presence/absence analysis containing a member from both a bilaterian and an outgroup), *Xenoturbella* and *Hofstenia* have very similar numbers (5459 and 5438, respectively). Gene, intron, and exon lengths all also fall within the range seen in many other invertebrate species (*Francis and Wörheide, 2017*). It thus seems that basic genomic features in *Xenoturbella* are not anomalous among Bilateria. Unlike some extremely simplified animals, such as orthonectids, we observe no extreme reduction in gene content.

All classes of homeodomain transcription factors have previously been reported to exist in Xenacoelomorpha (*Brauchle et al., 2018*). We have identified five HOX-genes in *X. bocki* and at least four, and probably all five of these are on one chromosomal scaffold within 187 Kbp. *X. bocki* also has the ParaHox genes Gsx and Cdx; while Xlox/pdx is not found, it is present in Cnidarians and must therefore have been lost (*Jimenez-Guri et al., 2006*). If block duplication models of Hox and ParaHox evolutionary relationships are correct, the presence of a complete set of ParaHox genes implies the existence of their Hox paralogs in the ancestor of Xenacoelomorphs, suggesting the xenacoelomorph ancestor also possessed a Hox 3 ortholog. If anthozoans also have an ortholog of bilaterian Hox 2 (*Ryan et al., 2006*), this must also have been lost from Xenacoelomorphs. The minimal number of Hox genes in the xenacoelomorph stem lineage was therefore probably 7 (AntHox1, lost Hox2, lost Hox 3, CentHox 1, CentHox 2, CentHox 3, and postHoxP).

Based on early sequencing technology and without a reference genome available, it was thought that Acoelomorpha lack many bilaterian microRNAs. Using deep sequencing of small RNAs and our high-quality genome, we have shown that *Xenoturbella* shows a near-complete bilaterian set of miRNAs including the single deuterostome-specific miRNA family (MIR-103) (*Figure 9*). The low number of differential family losses of *Xenoturbella* (8 of 31 bilaterian miRNA families) inferred is the same as the number lost in the flatworm *Schmidtea,* and substantially lower than the number lost in the rotifer *Brachionus* (which has lost 14 bilaterian families). It is worth mentioning that *X. bocki* shares the absence of one miRNA family (MIR-216) with all Ambulacrarians, although if Deuterostomia are paraphyletic this could be interpretable as a primitive state (*Kapli et al., 2021*).

The last decade has seen a re-evaluation of our understanding of the evolution of the neuropeptide signaling genes (*Jékely, 2013*; *Mirabeau and Joly, 2013*). The peptidergic systems are thought to have undergone a diversification that produced approximately 30 systems in the bilaterian common ancestor (*Jékely, 2013*; *Mirabeau and Joly, 2013*). Our study identified 31 neuropeptide systems in *X. bocki,* and for all of these either the ligand, receptor, or both are also present in both protostomes and deuterostomes, indicating conservation across Bilateria. It is likely that more ligands (which are short and variable) remain to be found with better detection methods. It appears that the *Xenoturbella* genome contains a nearly complete bilaterian neuropeptide complement with no signs of simplification but rather signs of expansions of certain gene families. Our analyses also reveal a potential synapomorphy linking Xenacoelomorpha with Ambulacraria (*Figure 8* and https://doi.org/10.5281/zenodo.6962271).

We have used the predicted presence and absence of genes across a selection of metazoan genomes as characters for phylogenetic analyses. Our trees reconfirm the findings of recent phylogenomic gene alignment studies in linking *Xenoturbella* to the Ambulacraria. We also used these data to test different bilaterians for their propensity to lose otherwise conserved genes (or for our inability to identify orthologs; *Natsidis et al., 2021*). While the degree of gene loss appears similar between *Xenoturbella* and acoels, the phylogenetic analysis shows longer branches leading to the acoels, most likely due to faster evolution, gain of lineage-specific genes, and some degree of gene loss in the branch leading to the Acoelomorpha. Recent work has shown the tendency of rapidly evolving genes (in particular those belonging to rapidly evolving species) to be missed by orthology detection software (*Natsidis et al., 2021*; *Weisman et al., 2020*).

This pattern of conservation of evolutionarily old parts of the Metazoan genome is further reinforced by the retention in *Xenoturbella* of linkage groups present from sponges to vertebrates. It is interesting to note that *X. bocki* does not follow the pattern seen in other morphologically simplified animals such as nematodes and platyhelminths, which have lost and/or fused these ALGs. We interpret this to be a signal of comparably slower genomic evolution in *Xenoturbella* in comparison to some other bilaterian lineages. The fragmented genome sequence of *Hofstenia* prevents us from asking whether the ancient linkage groups have also been preserved in the Acoelomorpha.

One of the chromosome-scale scaffolds in our assembly showed a different methylation and age signal, with both older and younger genes, and no clear relationship to metazoan linkage groups. By analyzing orthogroups of genes on this scaffold for their phylogenetic signal and finding *X. bocki* genes to cluster with those of *X. profunda,* we concluded that the scaffold most likely does not represent a contamination. It remains unclear whether this scaffold is a fast-evolving chromosome or a chromosomal fragment or arm. Very fast evolution on a chromosomal arm has, for example, been shown in the zebrafish (*Howe et al., 2016*).

Apart from DNA from *X. bocki*, we also obtained a highly contiguous genome of a species related to marine *Chlamydia* species (known from microscopy to exist in *X. bocki*); a symbiotic relationship between *Xenoturbella* and the bacterium has been thought possible (*Pillonel et al., 2018*; *Robertson et al., 2024*). The large gene number and the completeness of genetic pathways we found in the chlamydial genome do not support an endosymbiotic relationship.

Overall, we have shown that, while *Xenoturbella* has lost some genes – in addition to the reduced number of Hox genes previously noted, we observe a reduction of some signaling pathways to the core components – in general, the *X. bocki* genome is not strikingly simpler than many other bilaterian genomes. We do not find support for a strong version of the Nephrozoa hypothesis that would predict many missing bilaterian genes. Bilaterian Hox and microRNA absent from Acoelomorpha are found in *Xenoturbella* eliminating the impact of two character types that were previously cited in support of Nephrozoa. The *Xenoturbella* genome has also largely retained the ALGs found in other bilaterians and does not represent a structure intermediate between Eumetazoan and bilaterian ground states. Overall, while we can rule out a strong version of the Nephrozoa hypothesis with many Bilaterian characteristics missing in xenacoelomorphs, our analysis of the *Xenoturbella* genome cannot distinguish between a weak version of Nephrozoa and the Xenambulacraria topology.

## Materials and methods
### Genome sequencing, assembly, and scaffolding

We extracted DNA from individual *Xenoturbella* specimens with a standard and additionally worked with a Phenol–Chloroform protocol specifically developed to extract HMW DNA (dx.doi.org/10.17504/protocols.io.mrxc57n). The extracted DNA was quality controlled with a Nanodrop instrument in our laboratory and subsequently a TapeStation at the sequencing center. Worms were first starved and kept in repeatedly replaced salt water, reducing the likelihood of food or other contaminants in the DNA extractions. First, we sequenced Illumina short paired-end reads and mate pair libraries (see *Philippe et al., 2019* for details). As the initial paired-read datasets were of low complexity and coverage, we later complemented these data with an Illumina HiSeq 2000/2500 series paired-end dataset with ~700 bp insert size and 250 bp read lengths, yielding ~354 million reads. Additionally, we generated ~40 million Illumina TruSeq Synthetic Long Reads (TSLR) for high-confidence primary scaffolding.

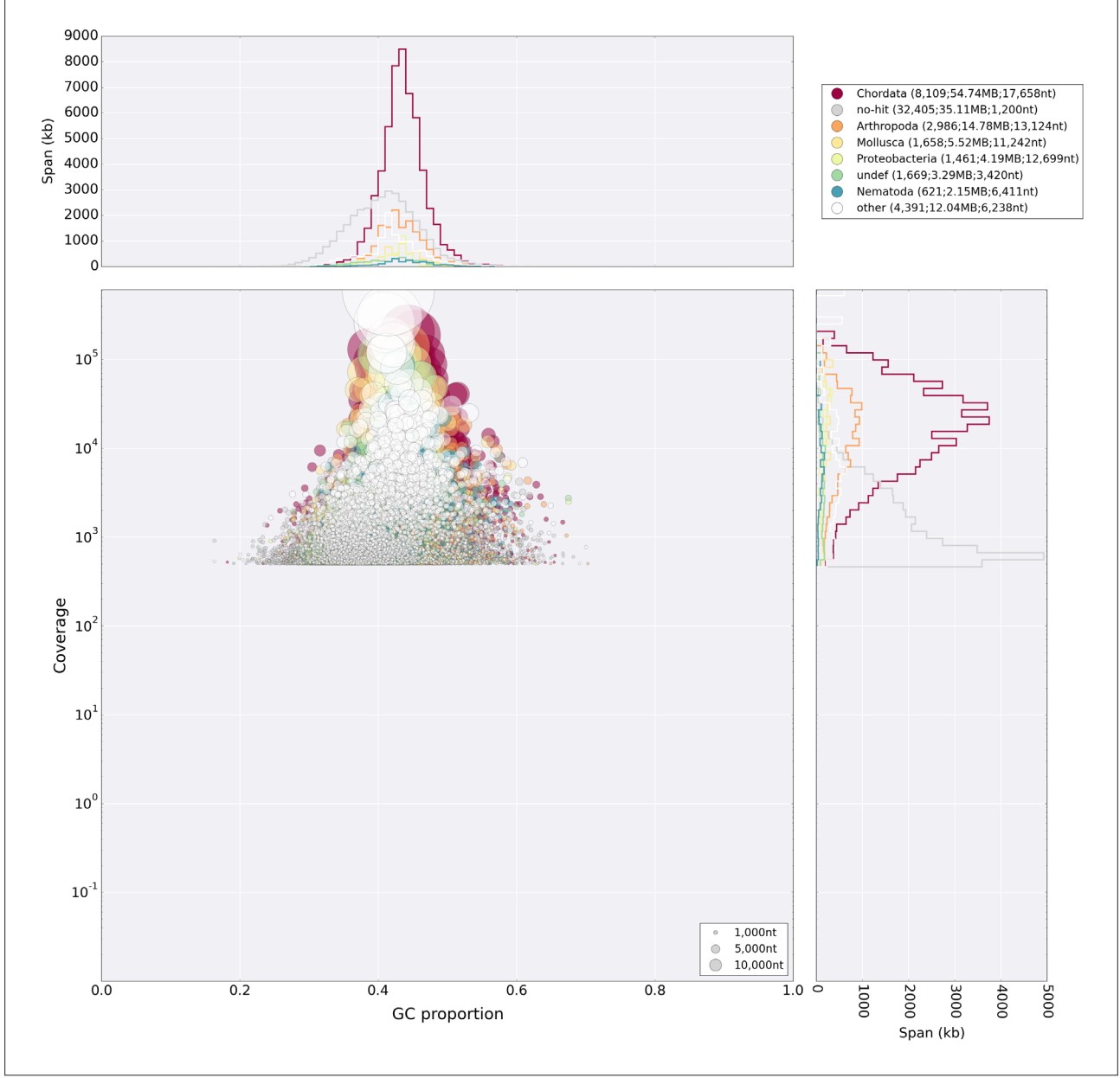

**Figure 12.** Blobplot analysis of the primary Illumina genome assembly. The assembly shows no major microorganismal contamination, apart from the *Chlamydia* and Gammaproteobacteria described in the main text. The diamond tool was used to blast against the UniProt database for this analysis.

After read cleaning with Trimmomatic v.0.38 (**Bolger et al., 2014**), we conducted initial test assemblies using the clc assembly cell v.5 and ran the blobtools pipeline (**Laetsch and Blaxter, 2017b**) to screen for contamination (**Figure 12**). Not detecting any significant numbers of reads from suspicious sources in the HiSeq dataset, we used SPAdes v. 3.9.0 (**Bankevich et al., 2012**) to correct and assemble a first draft genome. We also tried to use dipSPAdes but found the runtime to exceed several weeks without finishing. We submitted the SPAdes assembly to the redundans pipeline to eliminate duplicate contigs and to scaffold with all available mate pair libraries. The resulting assembly was then further scaffolded with the aid of assembled transcripts (see below) in the BADGER pipeline (**Elsworth et al., 2013**). In this way, we were able to obtain a draft genome with ~60 kb N50 that could be scaffolded to chromosome scale super-scaffolds with the use of 3C data.

We also used two remaining specimens to extract HMW DNA for Oxford Nanopore Technologies GridION sequencing in collaboration with the Loman Laboratory in Birmingham. Unfortunately, the

extraction failed for one individual, with the DNA appearing to be contaminated with a dark-colored residue. We were able to prepare a ligation and a PCR library for DNA from the second specimen and obtain some genomic data. However, due to pore blockage on both flow cells, the combined data amounted to only about 0.5-fold coverage of the genome and was thus not useful in scaffolding. We suspect that the dark coloration of the DNA indicates a natural modification to be present in *X. bocki* DNA that inhibits sequencing with the Oxford Nanopore method.

Library preparation for genome-wide bisulfite sequencing was performed as previously described (*Lewis et al., 2020*). The resulting sequencing data were aligned to the *X. bocki* draft genome using Bismark in non-directional mode to identify the percentage of methylation at each cytosine genome-wide. Only sites with >10 reads mapping were considered for further analysis.

## Preparation of the Hi-C libraries

The Hi-C protocol was adapted at the time from *Lieberman-Aiden et al., 2009*, *Sexton et al., 2012*, and *Marie-Nelly et al., 2014*. Briefly, an animal was chemically cross-linked for 1 hr at room temperature (RT) in 30 ml of PBS 1× added with 3% formaldehyde (Sigma – F8775 – 4 × 25 ml). Formaldehyde was quenched for 20 min at RT by adding 10 ml of 2.5 M glycine. The fixed animal was recovered through centrifugation and stored at –80°C until use. To prepare the proximity ligation library, the animal was transferred to a VK05 Precellys tubes in 1× DpnII buffer (New England Biolabs; 0.5 ml) and the tissues were disrupted using the Precellys Evolution homogenizer (Bertin-Instrument). SDS was added (0.3% final) to the lysate and the tubes were incubated at 65°C for 20 min, followed by an incubation at 37°C for 30 min and an incubation of 30 min after adding 50 µl of 20% triton-X100. A total of 150 units of the DpnII restriction enzyme were then added and the tubes were incubated overnight at 37°C. The endonuclease was inactivated 20 min at 65°C and the tubes were then centrifuged at 16,000 × *g* during 20 min, supernatant was discarded, and pellets were resuspended in 200 µl NE2 1× buffer and pooled. DNA ends were labeled using 50 µl NE2 10× buffer, 37.5 µl 0.4 mM dCTP-14-biotin, 4.5 µl 10 mM dATP-dGTP-dTTP mix, 10 µl klenow 5 U/µl, and incubation at 37°C for 45 min. The labeling mix was then transferred to ligation reaction tubes (1.6 ml ligation buffer; 160 µl ATP 100 mM; 160 µl BSA 10 mg/ml; 50 µl T4 DNA ligase [New England Biolabs, 5 U/µl]; 13.8 ml $H_2O$) and incubated at 16°C for 4 hr. A proteinase K mix was added to each tube and incubated overnight at 65°C. DNA was then extracted, purified, and processed for sequencing as previously described (*Baudry et al., 2020*). Hi-C libraries were sequenced on a NextSeq 500 (2 × 75 bp, paired-end using custom-made oligonucleotides as in *Marie-Nelly et al., 2014*). Libraries were prepared separately on two individuals in this way but eventually merged. Note that a more recent version of the HI-C protocol than the one used here has been described elsewhere (*Lafontaine et al., 2021*).

## instaGRAAL assembly preprocessing

The primary Illumina assembly contains a number of very short contigs, which are disruptive when computing the contact distribution needed for the instaGRAAL proximity ligation scaffolding (pre-release version, see *Marie-Nelly et al., 2014* and *Baudry et al., 2020* for details). Testing several Nx metrics, we found a relative length threshold that depends on the scaffolds' length distribution to be a good compromise between the need for a low-noise contact distribution and the aim of connecting most of the genome. We found N90 a suitable threshold and excluded contigs below 1308 bp. This also ensured no scaffolds shorter than three times the average length of a DpnII restriction fragment (RF) were in the assembly. In this way, every contig contained enough RFs for binning and were included in the scaffolding step.

Reads from both libraries were aligned with bowtie2 (v. 2.2.5) (*Langmead and Salzberg, 2012*) against the DpnII RFs of the reference assembly using the hicstuff pipeline (https://github.com/koszullab/hicstuff; *koszullab, 2018*) and in paired-end mode (with the options: -fg-maxins 5 -fg-very-sensitive-local), with a mapping quality >30. The preprocessed genome was reassembled using instaGRAAL. Briefly, the program uses a Markov Chain Monte Carlo method that samples DNA segments (or bins) of the assembly for their best relative 1D positions with respect to each other. The quality of the positions is assessed by fitting the contact data first on a simple polymer model, then on the plot of contact frequency according to the genomic distance law computed from the data. The best relative position of a DNA segment with respect to one of its most likely neighbors consists in operations such as flips, swaps, merges, or a split of contigs. Each operation is either accepted or rejected based

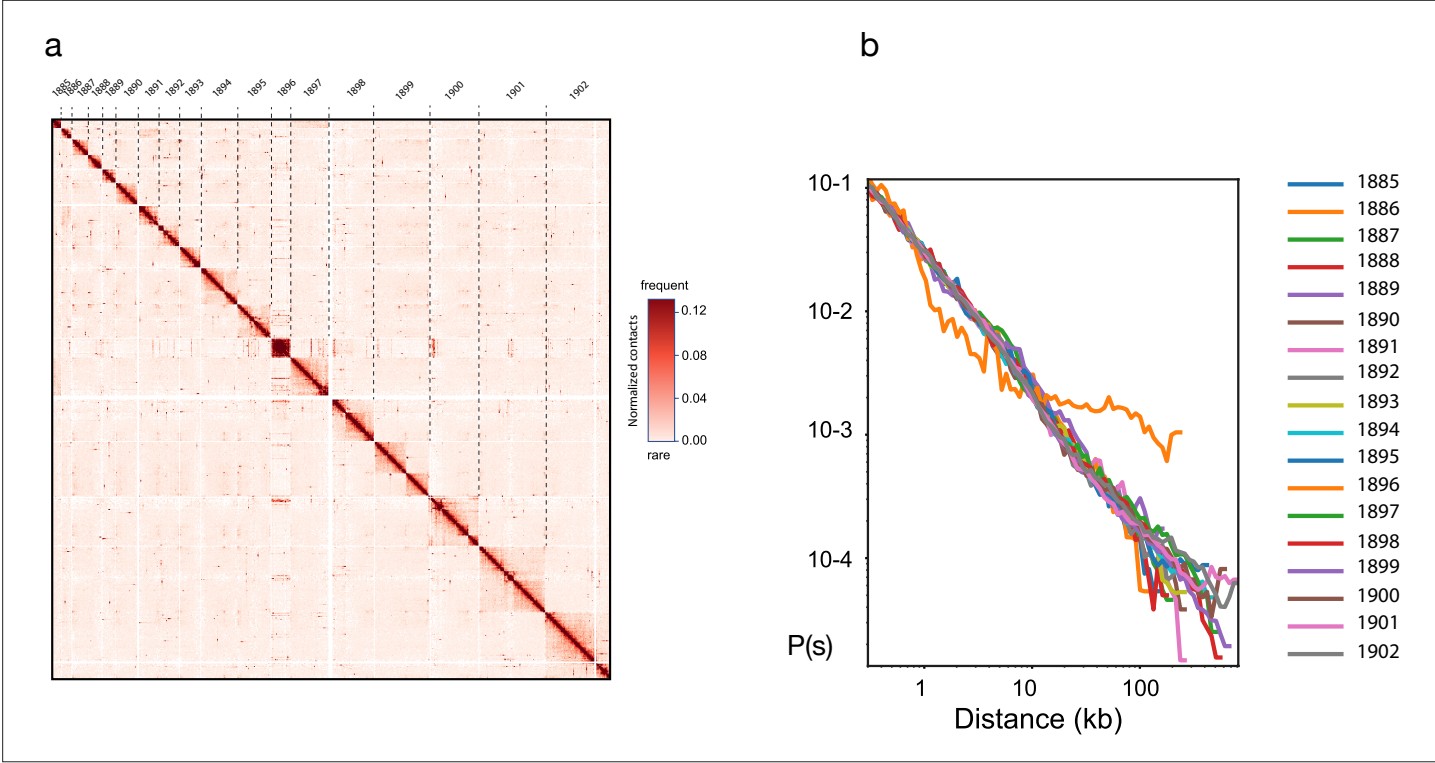

**Figure 13.** Hi-C based genome scaffolding with instaGRAAL. (**a**) Contact frequency map of the largest 18 scaffolds and (**b**) distribution of contact frequency as a function of distance (distance law).

The online version of this article includes the following figure supplement(s) for figure 13:

**Figure supplement 1.** Kmer profile of the *X. bocki* Illumina WGS reads obtained with GenomeScope2 (*Ranallo-Benavidez et al., 2020*).

on the computed likelihood, resulting in an iterative progression toward the 1D structure that best fits the contact data. Once the entire set of DNA segments is sampled for position (i.e., a cycle), the process starts over. The scaffolder was run independently for 50 cycles, long enough for the chromosome structure to converge. The corresponding genome is then considered stable and suitable for further analyses (*Figure 13*). The scaffolded assemblies were then refined using instaGRAAL's instaPolish module to correct small artifactual inversions that are sometimes a by-product of instaGRAAL's processing.

## Genome annotation

### Transcriptome sequencing

We extracted total RNA from a single *X. bocki* individual and sequenced a strand-specific Illumina paired-end library. Extraction of total RNA was performed using a modified Trizol & RNeasy hybrid protocol for which tissue had to be stored in RNAlater. cDNA transcription reaction/cDNA synthesis was done using the RETROscript kit (Ambion) using both Oligo(dT) and Random Decamer primers. Detailed extraction and transcription protocols are available from the corresponding authors. The resulting transcriptomic reads (deposited under SRX20415651) were assembled with the Trinity pipeline (*Haas et al., 2013*; *Trinity, 2015*) into 103,056 sequences (N50: 705; BUSCO_v5 Eukaryota scores: C: 65.1%, [S: 34.1%, D: 31.0%], F: 22.0%, M: 12.9%) for initial control and then supplied to the genome annotation pipeline (below).

### Repeat annotation

In the absence of a repeat library for Xenoturbellida, we first used RepeatModeller v. 1.73 to establish a library de novo. We then used RepeatMasker v. 4.1.0 (https://www.repeatmasker.org) and the Dfam library (*Wheeler et al., 2013*; *Hubley et al., 2016*) to soft-mask the genome. We mapped the repeats to the instaGRAAL scaffolded genome with RepeatMasker.

## Gene prediction and annotation

We predicted genes using AUGUSTUS (*Stanke and Waack, 2003*) implemented into the BRAKER (v.2.1.0) pipeline (*Hoff et al., 2019*; *Hoff et al., 2016*) to incorporate the RNA-seq data. BRAKER uses spliced aligned RNA-seq reads to improve training accuracy of the gene finder GeneMark-ET (*Lomsadze et al., 2014*). Subsequently, a highly reliable gene set predicted by GeneMark-ET in ab initio mode was selected to train the gene finder AUGUSTUS, which in a final step predicted genes with evidence from spliced aligned RNA-seq reads. To make use of additional single-cell transcriptome data allowing for a more precise prediction of 3'-UTRs, we employed a production version of BRAKER (August 2018 snapshot). We had previously mapped the RNA-seq data to the genome with gmap-gsnap v. 2018-07-04 (*Wu et al., 2016*) and used samtools (*Li et al., 2009*) and bamtools (*Barnett et al., 2011*) to create the necessary input files. This process was repeated in an iterative way, visually validating gene structures and comparing with mappings loci inferred from a set of single-cell RNA-seq data (published elsewhere, see *Robertson et al., 2022*) in particular regarding fused genes. Completeness of the gene predictions was independently assessed with BUSCO_v5 (*Simão et al., 2015*) setting the metazoan and the eukaryote datasets as reference respectively on gVolante (*Nishimura et al., 2017*). We used InterProScan v. 5.27-66.0 standalone (*Jones et al., 2014*; *Mulder and Apweiler, 2007*) on the UCL cluster to annotate the predicted *X. bocki* proteins with Pfam, SUPERFAM, PANTHER, and Gene3D information.

## Horizontal gene transfer

To detect horizontally acquired genes in the *X. bocki* genome, we used a pipeline available from https://github.com/reubwn/hgt (*Nowell, 2016*). Briefly, this uses blasts against the NCBI database, alignments with MAFFT (*Katoh and Standley, 2013*), and phylogenetic inferences with IQTREE (*Minh et al., 2020*; *Nguyen et al., 2015*) to infer most likely horizontally acquired genes, while trying to discard contamination (e.g., from co-sequenced gut microbiota).

## Orthology inference

We included 155 metazoan species and outgroups in our orthology analysis. We either downloaded available proteomes or sourced RNA-seq reads from online repositories to then use Trinity v 2.8.5 and Trinnotate v. 3.2.0 to predict protein sets. In the latter case, we implemented diamond v. 2.0.0 blast (*Buchfink et al., 2015*; *Buchfink et al., 2021*) searches against UniProt and Pfam (*Finn et al., 2016*) hmm screens against the Pfam-A dataset into the prediction process. We had initially acquired 185 datasets, but excluded some based on inferior BUSCO completeness, while at the same time aimed to span as many phyla as possible. Orthology was then inferred using OrthoFinder v. 2.2.7 (*Emms and Kelly, 2019*; *Emms and Kelly, 2015*), again with diamond as the blast engine.

Using InterProScan v. 5.27-66.0 standalone on all proteomes, we added functional annotation and then employed kinfin (*Laetsch and Blaxter, 2017a*) to summarize and analyze the orthology tables. For the kinfin analysis, we tested different query systems in regard to phylogenetic groupings (see supplementary material).

To screen for inflation and contraction of gene families, we first employed CAFE5 (*Han et al., 2013*), but found the analysis to suffer from long branches and sparse taxon sampling in Xenambulacraria. We thus chose to query individual gene families (e.g., transcription factors) by looking up Pfam annotations in the InterProScan tables of high-quality genomes in our analysis.

Through the GenomeMaple online platform, we calculated completeness of signaling pathways within the KEGG database using GhostX as the search engine.

## Presence/absence phylogenetics

We used a database of metazoan proteins, updated from *Leclère et al., 2018*, as the basis for an OMA analysis to calculate orthologous groups, performing two separate runs, one including *Xenoturbella* and acoels, and one with only *Xenoturbella*. We converted OMA gene OrthologousMatrix. txt files into binary gene presence absence matrices in Nexus format with datatype = restriction. We calculated phylogenetic trees on these matrices using RevBayes (see https://github.com/milliescient/metazoa-gene-content; *Walker, 2017* for RevBayes script), as described in *Mulder and Apweiler, 2007* with corrections for no absent sites and no singleton presence, using the reversible, not the

Dollo model, as it is more likely to be able to correct for noise related to prediction errors (*Pett et al., 2019*; *Buchfink et al., 2015*). For each matrix, two runs were performed and compared and consensus trees generated with bpcomp from Phylobayes (*Lartillot et al., 2009*).

## Hox and ParaHox gene cluster identification and characterization

Previous work has already used transcriptomic data and phylogenetic inference to identify the homeobox repertoire in *X. bocki*. These annotations were used to identify genomic positions and gene annotations that correspond to Hox and ParaHox clusters in *X. bocki*. Protein sequences of homeodomains (Evx, Cdx, Gsx, antHox1, centHox1, centHox2, cent3, and postHoxP) were used as TBLASTN queries to identify putative scaffolds associated with Hox and ParaHox clusters. Gene models from these scaffolds were compared to the full-length annotated homeobox transcripts from *Brauchle et al., 2018* using BLASTP, using hits over 95% identity for homeobox classification. There were some possible homeodomain-containing genes on the scaffolds that were not previously characterized and were therefore not given an annotation.

There were issues concerning the assignment of postHoxP and Evx to gene models. To ascertain possible CDS regions for these genes, RNA-seq reads were mapped with HISAT2 to the scaffold and to previous annotation (*Brauchle et al., 2018*), were assembled with Trinity, and these were combined with BRAKER annotations.

Some issues were also observed with homeodomain queries matching genomic sequences that were identical, suggesting artifactual duplications. To investigate contiguity around genes, the ONT reads were aligned with Minimap2 to capture long reads over regions and coverage.

## Small RNA sequencing and analysis

Two samples of starved worms were subjected to 5' monophosphate-dependent sequencing of RNAs between 15 and 36 nucleotides in length, according to previously described methods (*Sarkies et al., 2015*). Using miRTrace (*Kang et al., 2018*) 3.3, 18.6 million high-quality reads were extracted and merged with the 27,635 high-quality 454 sequencing reads from Philippe et al. The genome sequence was screened for conserved miRNA precursors using MirMachine (*Umu et al., 2022*), followed by a MirMiner run that used predicted precursors and processed and merged reads on the genome (*Wheeler et al., 2009*). Outputs of MirMachine and MirMiner were manually curated using a uniform system for the annotation of miRNA genes (*Fromm et al., 2015*) and by comparing to MirGeneDB (*Fromm et al., 2022*).

## Neuropeptide prediction and screen

Neuropeptide prediction was conducted on the full set of *X. bocki* predicted proteins using two strategies to detect neuropeptide sequence signatures. First, using a custom script detecting the occurrence of repeated sequence patterns: RRx(3,36)RRx(3,36)RRx(3,36)RR,RRx(2,35)ZRRx(2,35)ZRR, RRx(2,35)GRRx(2,35)GRR, RRx(1,34)ZGRRx(1,34)ZGRR where R = K or R; x = any amino acid; Z = any amino acid but repeated within the pattern. Second, using HMMER3.1 (*Johnson et al., 2010*) (http://www.hmmer.org/), and a combination of neuropeptide HMM models obtained from the PFAM database (http://pfam.xfam.org/) as well as a set of custom HMM models derived from the alignment of curated sets of neuropeptide sequences (*Jékely, 2013*; *Mirabeau and Joly, 2013*; *Zandawala et al., 2017*). Sequences retrieved using both methods and comprising fewer than 600 amino acids were further validated. First, by blast analysis: sequences with E-value ratio 'best blast hit versus ncbi nr database/best blast hit versus curated neuropeptide dataset' <1e-40 were discarded. Second by reciprocal best blast hit clustering using Clans (*Frickey and Lupas, 2004*) (https://www.bio.mpg.de/67908/clans) with a set of curated neuropeptide sequences (*Jékely, 2013*). SignalP-5.0 (*Almagro Armenteros et al., 2019*) (https://services.healthtech.dtu.dk/services/SignalP-5.0/) was used to detect the presence of a signal peptide in the curated list of predicted neuropeptide sequences while Neuropred (*Southey et al., 2006*) (http://stagbeetle.animal.uiuc.edu/cgi-bin/neuropred.py) was used to detect cleavage sites and post-translational modifications. Sequence homology of the predicted sequence with known groups was analyzed using a combination of (i) blast sequence similarity with known bilaterian neuropeptide sequences, (ii) reciprocal best blast hit clustering using Clans and sets of curated neuropeptide sequences, and (iii) phylogeny using MAFFT (https://mafft.cbrc.jp/alignment/server/), TrimAl (*Capella-Gutiérrez et al., 2009*) (https://trimal.cgenomics.org/), and IQ-TREE

(*Nguyen et al., 2015*) webserver for alignment, trimming, and phylogeny inference, respectively. Bilaterian Prokineticin-like sequences were searched in NCBI nucleotide, EST, and SRA databases as well as in the *Saccoglossus kowalevskii* genome assembly (*Nguyen et al., 2015*; *Simakov et al., 2015*) (https://groups.oist.jp/molgenu) using various bilaterian Prokineticin-related protein sequences as query. The sequences used for alignments shown in the figures were collected from the NCBI nucleotide and protein databases as well as from the following publications: 7B2 (*Jékely, 2013*); NucB2 (*Zandawala et al., 2017*); insulin (*Cherif-Feildel et al., 2019*); and Prokineticin (*Negri and Ferrara, 2018*; *Ericsson and Söderhäll, 2018*; *Thiel et al., 2018b*). Alignments for figures were created with Jalview (https://www.jalview.org/).

## Neuropeptide receptor search

Neuropeptide receptor sequences for rhodopsin-type GPCR, secretin-type GPCR, and tyrosine and serine/threonine kinase receptors were searched by running HMMER3.1 on the full set of *X. bocki* predicted proteins using the 7tm_1 (PF00001), 7tm_2 (PF00002), and PK_Tyr_Ser-Thr (PF07714) HMM models, respectively, which were obtained from the PFAM database (http://pfam.xfam.org/). Sequences above the significance threshold were then aligned with sequences from the curated dataset, trimmed, and phylogeny inference was conducted using same method as for the neuropeptide. A second alignment and phylogeny inference was conducted after the removal of all *X. bocki* sequences having no statistical support for grouping with any of the known neuropeptide receptors from the curated dataset. Curated datasets were collected from the following publications: rhodopsin-type GPCR beta and gamma and secretin-type GPCR (*Thiel et al., 2018b*); rhodopsin-type GPCR delta (leucine-rich repeat-containing G-protein-coupled receptors) (*Roch and Sherwood, 2014*); tyrosine kinase receptors (*de Oliveira et al., 2019*; *Smýkal et al., 2020*); and were complemented with sequences from the NCBI protein database.

## Synteny

Ancestral linkage analyses rely on mutual-best-hits computed using Mmseqs2 (*Steinegger and Söding, 2017*) between pairs of species in which chromosomal assignments to ALGs were previously performed, such as *Branchiostoma floridae* or *Pecten maximus* (*Simakov et al., 2020*). Oxford dotplots were computed by plotting reciprocal positions of indexed pairwise orthologs between two species as performed previously (*Simakov et al., 2020*; *Simakov et al., 2022*). The significance of ortholog enrichment in pairs of chromosomes was assessed using a Fisher test.

We also used a Python implementation of MCscanX (*Wang et al., 2012*; *Tang, 2010*) available on https://github.com/tanghaibao/jcvi/wiki/MCscan-(Python-version) to compare *X. bocki* to *E. muelleri*, *Trichoplax adhaerens*, *B. floridae*, *S. kowalevskii*, *Ciona intestinalis*, *Nematostella vectensis*, *Asteria rubens*, *P. maximus*, *Nemopilema nomurai*, and *Carcinoscorpius rotundicauda*. Briefly, the pipeline uses high-quality genomes and their annotations to infer syntenic blocks based on proximity. For this, an all vs. all blastp is performed and synteny extended from the anchors identified in this way. Corresponding heatmaps were plotted with Python in a Jupyter notebooks instance.

## *Chlamydia* assembly, annotation and phylogenetics

We identified a highly contiguous *Chlamydia* genome in the *X. bocki* genome assembly using blast. We then used our Oxford Nanopore-derived long reads to scaffold the *Chlamydia* genome with LINKS (*Warren et al., 2015*) and annotated it with the automated PROKKA pipeline. To place the genome on the *Chlamydia* tree, we extracted the 16S ribosomal RNA gene sequence, aligned it with set of *Chlamydia* 16S rRNA sequences from *Dharamshi et al., 2020* using MAFFT, and reconstructed the phylogeny using IQ-TREE 2 (*Minh et al., 2020*) We visualized the resulting tree with Figtree (http://tree.bio.ed.ac.uk/).

# Acknowledgements

We thank Josh Quick and Nick Loman for help with the generation of ONT long-read data. Analyses were conducted mainly on the UCL Cluster, with some computations also run on the CHEOPS cluster at the University of Cologne. We are grateful to Kevin J Peterson for his comments on the manuscript, the miRNA section in particular. We thank the Kristineberg Center for Marine Research and Innovation

for their essential support in sampling *Xenoturbella*. PHS was funded by an ERC grant (ERC-2012-AdG 322790) to MJT, which also supported HR, ACZ, and SM. PHS was also funded through an Emmy-Noether grant (434028868) to himself. Part of this work was funded by BBSRC grant BB/R016240/1 (MJT/PK), a Leverhulme Trust Research Project Grant RPG-2018-302 (MJT/DJL), and the European Union's Horizon 2020 research and innovation program under the Marie Skłodowska-Curie grant agreement no 764840 IGNITE (MJT/PN).

## Additional information

### Funding

| Funder | Grant reference number | Author |
| --- | --- | --- |
| European Research Council | ERC-2012-AdG 322790 | Philipp H Schiffer<br>Helen E Robertson<br>Anne C Zakrzewski<br>Steven Müller<br>Maximilian J Telford |
| Deutsche Forschungsgemeinschaft | 434028868 | Philipp H Schiffer |
| Biotechnology and Biological Sciences Research Council | BB/R016240/1 | Paschalis Natsidis<br>Maximilian J Telford |
| Leverhulme Trust | RPG-2018-302 | Daniel J Leite<br>Maximilian J Telford |
| HORIZON EUROPE Marie Sklodowska-Curie Actions | 10.3030/764840 | Paschalis Natsidis<br>Maximilian J Telford |

The funders had no role in study design, data collection and interpretation, or the decision to submit the work for publication.

### Author contributions

Philipp H Schiffer, Conceptualization, Data curation, Software, Formal analysis, Supervision, Investigation, Visualization, Methodology, Writing – original draft, Project administration, Writing – review and editing; Paschalis Natsidis, Data curation, Software, Formal analysis, Visualization; Daniel J Leite, Formal analysis, Visualization, Writing – original draft, Writing – review and editing; Helen E Robertson, Formal analysis, Investigation; François Lapraz, Ferdinand Marlétaz, Data curation, Formal analysis, Visualization, Writing – original draft, Writing – review and editing; Bastian Fromm, Data curation, Formal analysis, Visualization, Writing – original draft; Liam Baudry, Data curation, Writing – original draft; Fraser Simpson, Methodology; Eirik Høye, Data curation, Visualization; Anne C Zakrzewski, Formal analysis, Visualization; Paschalia Kapli, Formal analysis; Katharina J Hoff, Software, Writing – review and editing; Steven Müller, Data curation, Formal analysis; Martial Marbouty, Data curation, Methodology; Heather Marlow, Supervision; Richard R Copley, Conceptualization, Formal analysis, Writing – original draft, Writing – review and editing; Romain Koszul, Formal analysis, Supervision, Writing – original draft, Writing – review and editing; Peter Sarkies, Formal analysis, Methodology, Writing – original draft, Writing – review and editing; Maximilian J Telford, Conceptualization, Supervision, Methodology, Writing – original draft, Project administration, Writing – review and editing

### Author ORCIDs

Philipp H Schiffer ⓘ https://orcid.org/0000-0001-6776-0934
Daniel J Leite ⓘ https://orcid.org/0000-0001-9966-4760
François Lapraz ⓘ https://orcid.org/0000-0001-9209-2018
Bastian Fromm ⓘ https://orcid.org/0000-0003-0352-3037
Martial Marbouty ⓘ https://orcid.org/0000-0002-1668-8423
Romain Koszul ⓘ https://orcid.org/0000-0002-3086-1173
Peter Sarkies ⓘ https://orcid.org/0000-0003-0279-6199
Maximilian J Telford ⓘ https://orcid.org/0000-0002-3749-5620

Decision letter and Author response
Decision letter https://doi.org/10.7554/eLife.94948.sa1
Author response https://doi.org/10.7554/eLife.94948.sa2

## Additional files

### Supplementary files

• Supplementary file 1. Excel table with data sources for OrthoFinder analysis.

• Supplementary file 2. Full tree representation of the sequence similarities analysis of bilaterian insulin-related peptides. Tree is calculated from concatenated alignment of A and B chains. Numbers represent support for nodes calculated using 1000 ultrafast bootstrap replications and 1000 SH-aLRT replicates, respectively. Scale bar unit for branch length is the number of substitutions per site. Branches are colored according to the phylogenetic position of the organism from which the sequence originates: red, Xenoturbella; pink, Ambulacraria; blue, Chordata; orange, Ecdysozoa; green, Ecdysozoa; gray, Cnidaria. dILP, Drosophila insulin-like peptide; GSS, gonad-stimulating substance; ILP, insulin-like peptide; IGF, insulin-like growth factor. Radial version of this tree is presented in *Figure 8—figure supplement 2*. Sequences are available as *Figure 8—source data 1*; alignment and IQTREE tree files are available at https://doi.org/10.5281/zenodo.6962271.

• Supplementary file 3. Full tree representation of the phylogenetic analysis of bilaterian Leucine-rich repeat-containing G-protein coupled Receptors (Rhodopsin type G-protein coupled Receptors delta). Numbers represent support for nodes calculated using 1000 Ultrafast bootstrap replications and 1000 SH-aLRT replicates respectively. Scale bar unit for branch length is the number of substitutions per site. Branches are colored according to the phylogenetic position of the organism from which the sequence originates: red, Xenoturbella; pink, Ambulacraria; blue, Chordata; orange, Ecdysozoa; green, Ecdysozoa; gray, Cnidaria. Collapsed group colored in red indicate that they contain at least one *X. bocki* sequence. GPA2, Glycoprotein Hormone alpha5; GPB5, Glycoprotein Hormone beta2; GPCR, G Protein-Coupled Receptor; GRL-101, G-protein coupled receptor GRL101. Circular version of this tree is presented in *Figure 8—figure supplement 3*. Sequences are available as *Figure 8—source data 2*; alignment and IQTREE tree files are available at https://doi.org/10.5281/zenodo.6962271.

• Supplementary file 4. Full tree representation of the phylogenetic analysis of bilaterian Rhodopsin type G-protein coupled Receptors beta and gamma. Numbers represent support for nodes calculated using 1000 ultrafast bootstrap replications and 1000 SH-aLRT replicates respectively. Scale bar unit for branch length is the number of substitutions per site. Branches are colored according to the phylogenetic position of the organism from which the sequence originates: red, Xenoturbella; pink, Ambulacraria; blue, Chordata; orange, Ecdysozoa; green, Ecdysozoa; gray, Cnidaria. White boxes with associated name highlight groups of annotated sequences. AKH, adipokinetic hormone; Asta-A, Allatostatin-A; Asta-C, Allatostatin-C; CAPA, Cardio acceleratory peptide; CCAP, crustacean cardioactive peptide; CCHa, CCHamide peptide; CCK, cholecystokinin; CRZ, Corazonin; eFMRF, ecdysozoan-FMRFamide peptide; GGN-EP, GGN excitatory peptide; ETH, ecdysis triggering hormone; GnRH, Gonadotropin Releasing Hormone; GPR150, G Protein-Coupled Receptor 150; GPR54, G Protein-Coupled Receptor 54; GPR83, G Protein-Coupled Receptor 83; MCH, melanin concentrating hormone; NK-2, Neurokinin 2; Np-B/W, Neuropeptide B/W; Np-FF, Neuropeptide FF; Np-F, Neuropeptide F; Np-S, Neuropeptide S; Np-Y, Neuropeptide Y; PBAN, pheromone biosynthesis activation neuropeptide; PEN, neuroendocrine peptide PEN; PRP, Prolactin releasing peptide; QRFP, Neuropeptide QRFP; RYa, RYamide peptide; SIFa, SIFamide peptide; SPR, Sex peptide receptor; tFMRFa, trochozoan-FMRFamide peptide; TRH, thyrotrophin-releasing hormone. Circular version of this tree is presented in *Figure 8—figure supplement 4*. Sequences are available as *Figure 8—source data 2*; alignment and IQTREE tree files are available at https://doi.org/10.5281/zenodo.6962271.

• Supplementary file 5. Full tree representation of the phylogenetic analysis of bilaterian Tyrosine kinase Receptors. Numbers represent support for nodes calculated using 1000 ultrafast bootstrap replications and 1000 SH-aLRT replicates respectively. Scale bar unit for branch length is the number of substitutions per site. Branches are colored according to the phylogenetic position of the organism from which the sequence originates: red, Xenoturbella; pink, Ambulacraria; blue, Chordata; orange, Ecdysozoa; green, Ecdysozoa; gray, Cnidaria. Collapsed group colored in red indicate that they contain at least one *X. bocki* sequence. EGF, Epidermal Growth Factor;Discoidin cont. R, discoidin domain-containing receptor; Orphan Tyr. Kinase Ror2, receptor tyrosine kinase-

like orphan receptor 2; VKR, Venus kinase Receptor; ILP, Insulin-like peptide; PDGF, Platelet-derived growth factor; VEGF, Vascular endothelial growth factor; GDNF, Glial cell line-derived neurotrophic factor; FGF, fibroblast growth factor; PTTH, Prothoracicotropic hormone. Circular version of this tree is presented in *Figure 8—figure supplement 5*. Sequences are available as *Figure 8—source data 2*; alignment and IQTREE tree files are available at https://doi.org/10.5281/zenodo.6962271.

• Supplementary file 6. Full tree representation of the phylogenetic analysis of bilaterian Secretin type G-protein coupled Receptors. Numbers represent support for nodes calculated using 1000 ultrafast bootstrap replications and 1000 SH-aLRT replicates respectively. Scale bar unit for branch length is the number of substitutions per site. Branches are colored according to the phylogenetic position of the organism from which the sequence originates: red, Xenoturbella; pink, Ambulacraria; blue, Chordata; orange, Ecdysozoa; green, Ecdysozoa; gray, Cnidaria. White boxes with associated name highlight groups of annotated sequences. DH31,diuretic hormone 31; Np-RB1, Neuropeptide receptor B3; Np-RB4, Neuropeptide receptor B1; PDF, Pigment-dispersing factor; CRF, Corticotropin-releasingfactor; DH-44,diuretic hormone 44; PTH2/3-R,Parathyroid hormonereceptor2/3; GIP, Gastric inhibitory polypeptide; PACAP, Pituitary adenylate cyclase-activating polypeptide;VIP-R,Vasoactive intestinal polypeptide receptor; GHRH, Growth hormone-releasing hormone; PTH, Parathyroid hormone receptor; SCTR, Secretin Receptor. Circular version of this tree is presented in *Figure 8—figure supplement 6*. Sequences are available as *Figure 8—source data 2*; alignment and IQTREE tree files are available at https://doi.org/10.5281/zenodo.6962271.

• Supplementary file 7. Sequence alignment of bilaterian 7B2 neuropeptides. The alignment highlights the presence in all sequence of aconserved 'PPNPCP' motif. *X. bocki* sequence is highlighted by a red dashed line. Sequences are available as *Figure 8—source data 1*; alignment is available at https://doi.org/10.5281/zenodo.6962271.

• Supplementary file 8. Sequence alignment of Xenacoelomorpha LRIGamide neuropeptides. *X. bocki* sequence is highlighted by a reddashed line. Sequences are available as *Figure 8—source data 1*; alignment is available at https://doi.org/10.5281/zenodo.6962271.

• Supplementary file 9. Sequence alignment of bilaterian Nucleobindin2/Nesfatin neuropeptides. *X. bocki* sequence is highlighted by a reddashed line. Sequences are available as *Figure 8—source data 1*; alignment is available at https://doi.org/10.5281/zenodo.6962271.

• Supplementary file 10. Sequence alignment of Ambulacrarian octinsulin with a potential *X. bocki* homolog sequence. Red trianglehighlights the conserved cysteine positions. *X. bocki* sequence is highlighted by a red dashed line. Sequences are available as *Figure 8—source data 1*; alignment is available at https://doi.org/10.5281/zenodo.6962271.

• MDAR checklist

### Data availability

All read sets (RNA and DNA derived) used in this study will be made available with the publication of this manuscript on the SRA database under the BioProject ID PRJNA864813. Hi-C reads are deposited under SAMN30224387, RNA-Seq under SAMN35083895. The genome assemblies of X. bocki (ERS12565994, ERA16814408) and the Chlamydia sp. (ERS12566084, ERA16814775) are deposited under PRJEB55230 at ENA. Supplementary online material (described in the manuscript) has been made available on Zenodo.

The following datasets were generated:

| Author(s) | Year | Dataset title | Dataset URL | Database and Identifier |
|---|---|---|---|---|
| Schiffer P, Lapraz F | 2024 | Supplementary files for the *Xenoturbella bocki* genome analysis | https://doi.org/10.5281/zenodo.6962271 | Zenodo, 10.5281/zenodo.6962271 |
| Schiffer P, Lapraz F | 2022 | The genome of *Xenoturbella bocki* | https://www.ncbi.nlm.nih.gov/bioproject/?term=PRJNA864813 | NCBI BioProject, PRJNA864813 |
| Schiffer P, Lapraz F | 2022 | The slow evolving genome of the xenacoelomorph worm *Xenoturbella bocki* | https://www.ebi.ac.uk/ena/browser/view/PRJEB55230 | EBI European Nucleotide Archive, PRJEB55230 |

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
