## [Editor Report]

The authors provide a high-quality genome of the xenacoelomorph worm *Xenoturbella bocki* and discuss its structure and evolution. Understanding the genomic structure of this group provides important insights into bilaterian evolution. The authors make a solid case that the data they present is consistent with Xenacoelomorpha being a secondarily simplified member of Deuterostomia rather than a primitively simple sister group to all other bilaterians.

---

## [Decision Letter]

**Decision letter after peer review:**

[Editors’ note: this paper had been reviewed by PCI Genomics: Fernandez, R. (2023) Genomic idiosyncrasies of *Xenoturbella bocki*: morphologically simple yet genetically complex. *Peer Community in Genomics,* 100235. 10.24072/pci.genomics.100235]

Thank you for submitting your article "The slowly evolving genome of the xenacoelomorph worm *Xenoturbella bocki*" for consideration by *eLife*. Your article has been reviewed by 2 peer reviewers, one of whom is a member of our Board of Reviewing Editors, and the evaluation has been overseen by Alan Moses as the Senior Editor.

The reviewers have discussed their reviews with one another, and the Reviewing Editor has drafted this to help you prepare a revised submission. The reviewing editor also took into account previous reviews from an online peer-review platform.

Essential revisions (for the authors):

1) Since the manuscript has already gone through several rounds of review it is relatively clear of errors and technical problems. However, the reviews have also apparently led to a loss of clarity and flow in the manuscript. We ask the authors to read through the manuscript and think about improvements to structure and flow to make it easier to read and to make the main message clearer.

2) Please increase the size and clarity of the figure labels.

3) Finally, the most important point, which was raised by reviewers in previous rounds of review and we feel was not addressed sufficiently, is the question of the non-monophyly of Xenacoelomorpha that emerges from the presence/absence analysis. This result potentially undermines the authors' main conclusions and message. The manuscript brushes this result aside without a serious discussion or well-supported argument. It is crucial that this be corrected before the manuscript can be accepted for publication.

*Reviewer #1 (Recommendations for the authors):*

I believe in the aphorism that Perfect is the enemy of Good. There are probably places where the text can be tightened, where statements can be more general and where ideas can be explained better, but I don't feel that any of these are necessary.

*Reviewer #2 (Recommendations for the authors):*

Synopsis: The manuscript describes the genome assembly and analysis of *Xenoturbella bocki*, a worm that bears many morphological features ascribed to basal bilateria. The authors aim to analyse this genome in an attempt to determine the phylogenetic position of *X. bocki* as a representative of Xenacoelomorpha and its associated acoels. In doing so, they want to inform the debate as to whether xenacoelomorph belong among, or is in fact paraphyletic to all bilaterians. The authors amassed evidence from gene presence/absence matrices, HOX and parahox complements, miRNA, neuropeptide repertoire and synteny to argue that *X. bocki*, thus Xenacoelomorpha is a typical bilaterian from the genomic perspective and most likely a sister group to Ambulacraria. Importantly, due to the discordant position of other related acoels, the authors stop short of completely rejecting the alternative hypothesis that places Xenacoelomorph as the outgroup to Nephrozoa, a group containing all protostomes and deuterostomes. To reconcile the incongruent placements between otherwise closely related species, the authors argue that *X. bocki* is representative owing to its slow turn-over in its genome.

General comments:

This paper is primarily structured as a description of the *Xenoturbella bocki* genome. From a technical point of view, the genome is a high-quality assembly, with a high contiguity and completeness. The logic of the description is clear and reasonably compelling. However, there is a broader underlying issue with the flow: the paper reads like it has been "retrofitted" to address the broader question of the position of xenacoelomorph. As a result, the paper reads somewhat like a collage of different text fragments and lacks unity. In part perhaps because the authors do not feel strongly about staking their claims stronger, defaulting to "slowly evolving genome" as a general summary of the paper also does not seem particularly strongly supported. In its current form, I am afraid this paper is not sufficiently strong for it to be accepted at *eLife*.

I will include some comments below to help the authors improve the manuscript.

Phylogenetic position of Xenacoelomorpha – main theme vs. side-point?

Based on the Abstract and Introduction, one would expect the phylogenetic position of Xenacoelomorpha to be the major driving question of this paper. Somewhat incongruously, this question seems not to feature centrally across the figure. The only place where this is addressed directly and analytically is in Figure 2b vs. 2c, with only very little details given. The attempt to reconcile the two contrasting results also feel somewhat off-hand, amounting to "artefact due to very fast evolution in this taxon" (Figure legend, Figure 2c; also L. 197-199). Besides that, all the other key sections seem to be a straight-forward presentation of *X. bocki* as having a regular bilaterian genome. Admittedly, this presentation lends support to the view that *X. bocki* is not paraphyletic to Nephrozoa. However, the focal test seems to be a. somewhat inconclusively supported by the discrepancy between *X. bocki* and *P. naikaniensis* and *H. miamia*, and b. something of an after-thought. Crucially, in Figure 3, these two other species are included in the analysis. However, it does not appear that there is a formal attempt at clustering the species based on the presence/absence of the gene family members. Instead, the cladograms were schematic cladograms in b/c drawn by the authors". This both seems to be a major oversight, or perhaps it belies the notion that the paper is set out to determine the phylogenetic position of Xenacoelomorpha using the *X. bocki* genome. In the rest of the text, these two genomes seem not to feature much in further discussion.

---

## [Author Response]

Essential revisions (for the authors):1) Since the manuscript has already gone through several rounds of review it is relatively clear of errors and technical problems. However, the reviews have also apparently led to a loss of clarity and flow in the manuscript. We ask the authors to read through the manuscript and think about improvements to structure and flow to make it easier to read and to make the main message clearer.2) Please increase the size and clarity of the figure labels.3) Finally, the most important point, which was raised by reviewers in previous rounds of review and we feel was not addressed sufficiently, is the question of the non-monophyly of Xenacoelomorpha that emerges from the presence/absence analysis. This result potentially undermines the authors' main conclusions and message. The manuscript brushes this result aside without a serious discussion or well-supported argument. It is crucial that this be corrected before the manuscript can be accepted for publication.

We have addressed the points above. Specifically, in regard to

1) we have modified the manuscript in a way to put more emphasis on the biology and evolution of Xenoturbella, reducing the discussion of phylogenetics aspects. We think this improves the flow of the manuscript;

We have worked on the figure labels for point 2);

In regard to point 3 we have also added additional content to the Results and the Discussion session. However, we feel that the argument of the reviewer works both ways round, as there is no well-supported argument for a paraphyly or polyphyly of Xenacoelomorpha. This includes papers supporting the Nephrozoa hypothesis. For example Cannon et al. 2016 do retrieve Xenacoelomorpha and their support for Nephrozoa becomes weaker, when they exclude Acoels from their analysis (see their Extended Data Figure 3).

Further replies to the authors comments are given point-by-point below.

Reviewer #1 (Recommendations for the authors):I believe in the aphorism that Perfect is the enemy of Good. There are probably places where the text can be tightened, where statements can be more general and where ideas can be explained better, but I don't feel that any of these are necessary.

We agree that the text of our manuscript has slightly suffered during previous rounds of revisions. We thus tried our best to condense some parts and explain the general ideas better, specifically in the Abstract, Introduction, and in parts of the Discussion.

Reviewer #2 (Recommendations for the authors):Synopsis: The manuscript describes the genome assembly and analysis of *Xenoturbella bocki*, a worm that bears many morphological features ascribed to basal bilateria. The authors aim to analyse this genome in an attempt to determine the phylogenetic position of *X. bocki* as a representative of Xenacoelomorpha and its associated acoels. In doing so, they want to inform the debate as to whether xenacoelomorph belong among, or is in fact paraphyletic to all bilaterians. The authors amassed evidence from gene presence/absence matrices, HOX and parahox complements, miRNA, neuropeptide repertoire and synteny to argue that *X. bocki*, thus Xenacoelomorpha is a typical bilaterian from the genomic perspective and most likely a sister group to Ambulacraria. Importantly, due to the discordant position of other related acoels, the authors stop short of completely rejecting the alternative hypothesis that places Xenacoelomorph as the outgroup to Nephrozoa, a group containing all protostomes and deuterostomes. To reconcile the incongruent placements between otherwise closely related species, the authors argue that *X. bocki* is representative owing to its slow turn-over in its genome.General comments:This paper is primarily structured as a description of the *Xenoturbella bocki* genome. From a technical point of view, the genome is a high-quality assembly, with a high contiguity and completeness. The logic of the description is clear and reasonably compelling. However, there is a broader underlying issue with the flow: the paper reads like it has been "retrofitted" to address the broader question of the position of xenacoelomorph. As a result, the paper reads somewhat like a collage of different text fragments and lacks unity. In part perhaps because the authors do not feel strongly about staking their claims stronger, defaulting to "slowly evolving genome" as a general summary of the paper also does not seem particularly strongly supported. In its current form, I am afraid this paper is not sufficiently strong for it to be accepted at eLife.

We do not agree on the statement that our manuscript has been retrofitted to address the broader question of the position position of the xenacoelomorpha. We are engaging with various questions on the biology and evolution of Xenoturbella, which are found in the current literature. As stated above, the phylogenetic position is a very important part of this discussion and we thus touch upon it repeatedly in the text. A significant number of the papers discussing these groups focus on their contested phylogenetic position. Reviewer 1 points out, the perfect is the enemy of the good and we fully agree that parts of our text needed to be “smoothened”. Thanks to the input of both Reviewers we think this has now been achieved.

I will include some comments below to help the authors improve the manuscript.Phylogenetic position of Xenacoelomorpha – main theme vs. side-point?Based on the Abstract and Introduction, one would expect the phylogenetic position of Xenacoelomorpha to be the major driving question of this paper. Somewhat incongruously, this question seems not to feature centrally across the figure. The only place where this is addressed directly and analytically is in Figure 2b vs. 2c, with only very little details given. The attempt to reconcile the two contrasting results also feel somewhat off-hand, amounting to "artefact due to very fast evolution in this taxon" (Figure legend, Figure 2c; also L. 197-199). Besides that, all the other key sections seem to be a straight-forward presentation of *X. bocki* as having a regular bilaterian genome. Admittedly, this presentation lends support to the view that *X. bocki* is not paraphyletic to Nephrozoa. However, the focal test seems to be a. somewhat inconclusively supported by the discrepancy between *X. bocki* and *P. naikaniensis* and *H. miamia*, and b. something of an after-thought. Crucially, in Figure 3, these two other species are included in the analysis. However, it does not appear that there is a formal attempt at clustering the species based on the presence/absence of the gene family members. Instead, the cladograms were schematic cladograms in b/c drawn by the authors". This both seems to be a major oversight, or perhaps it belies the notion that the paper is set out to determine the phylogenetic position of Xenacoelomorpha using the *X. bocki* genome. In the rest of the text, these two genomes seem not to feature much in further discussion

We have modified the Abstract and Introduction to clearly state that our manuscript uses the genome of *X. bocki* to learn more about the biology and evolution of an enigmatic marine animal, with an intriguing phylogenetic position, regardless whether this is at the base of the Bilateria or within the Deuterostomes.

Both acoel genomes were only added to the analysis in Figure 3 by request of a reviewer in a previous review. We had chosen to omit these, as well as our own data from Paratomella rubra (see main text), from most of our analyses, as the genomes are not of high enough quality. We fully agree that an analysis of acoel genomes in a phylogenetic framework is urgently needed, however, for this it will first be necessary to assemble several such genomes to chromosome level.